# Research Progress of Macromolecules in the Prevention and Treatment of Sepsis

**DOI:** 10.3390/ijms241613017

**Published:** 2023-08-21

**Authors:** Jingqian Su, Shun Wu, Fen Zhou, Zhiyong Tong

**Affiliations:** Fujian Key Laboratory of Innate Immune Biology, Biomedical Research Center of South China, College of Life Science, Fujian Normal University, Fuzhou 350117, China; qsx20221412@student.fjnu.edu.cn (S.W.); 15859446856@163.com (F.Z.); tongzhiyong1998@163.com (Z.T.)

**Keywords:** antibody, biomolecule, polypeptide, polysaccharide, protein, RNA, sepsis

## Abstract

Sepsis is associated with high rates of mortality in the intensive care unit and accompanied by systemic inflammatory reactions, secondary infections, and multiple organ failure. Biological macromolecules are drugs produced using modern biotechnology to prevent or treat diseases. Indeed, antithrombin, antimicrobial peptides, interleukins, antibodies, nucleic acids, and lentinan have been used to prevent and treat sepsis. In vitro, biological macromolecules can significantly ameliorate the inflammatory response, apoptosis, and multiple organ failure caused by sepsis. Several biological macromolecules have entered clinical trials. This review summarizes the sources, efficacy, mechanism of action, and research progress of macromolecular drugs used in the prevention and treatment of sepsis.

## 1. Introduction

Sepsis is a life-threatening organ dysfunction caused by a host response disorder resulting from infection [1]. It has a high incidence among patients in intensive care units (ICUs) and is one of the leading causes of death in such units. There are approximately 50 million reported cases of patients with sepsis worldwide every year, with a mortality rate of >20%; in 2017, approximately 11 million individuals died due to sepsis. Sepsis is often accompanied by persistent systemic inflammation, resulting in immune imbalance and damage and dysfunction of multiple organs, such as the lungs, liver, kidneys, and brain. Currently, there is no specific drug for the treatment of sepsis. In clinical practice, patients with infection or suspected infection with a Sequential Organ Failure Assessment (SOFA) score of ≥2 points are diagnosed with sepsis, and antibacterial drug treatment should be initiated within 3 h. Obvious infection sites may require surgical management. Anti-inflammatory and immunomodulatory drugs are used to improve the inflammatory response in patients. For patients with disseminated intravascular coagulation (DIC), timely anticoagulant therapy is necessary. Mechanical ventilation and other supportive treatments are required for patients with acute respiratory distress syndrome. Fluid resuscitation should be performed for shock patients. Vasoactive drugs are administered to maintain a mean arterial pressure ≥ 65 mmHg [1,2,3]. Currently, both large and small molecules have been used for the treatment and prevention of sepsis, such as human serum albumin (HSA) and antibiotics. Additionally, laboratory research suggests that afelimomab, fucoxanthin, and the fermented drink kombucha significantly improve sepsis-induced symptoms, particularly by reducing the inflammatory response [3,4,5].

Macromolecular drugs (molecular weight ≥ 1000 Da), also known as biological products, are biological macromolecules produced using modern biotechnology for the prevention or treatment of diseases. The Food and Drug Administration (FDA) categorizes macromolecular drugs into vaccines, blood products, and gene and cell therapy agents based on the nature of drugs. These drugs can include proteins, polypeptides, polysaccharides, antibodies, and nucleic acids. In comparison with small molecule drugs, macromolecular drugs offer advantages, such as precise targeting, high efficiency, and long half-life [6,7].

To date, summary articles on the application of macromolecules in sepsis are lacking. Therefore, this review focuses on articles published in PubMed during 2012–2023 that discuss the use of macromolecules in the treatment of sepsis and classify them according to their properties, therapeutic effects, and mechanisms of action. The inflammation detection methods described in these articles involve quantitative polymerase chain reaction and enzyme-linked immunosorbent assay technologies to measure the mRNA and protein levels of cytokines, such as interleukin (IL)-1β, IL-6, and tumor necrosis factor (TNF)-α, among others. Table 1 is a summary of the macromolecules in the full text, as follows:

## 2. Proteins and Peptides

### 2.1. Anticoagulant

Anticoagulant drugs are used to treat thrombotic diseases and can be divided into antiplatelet and anticoagulant drugs; anticoagulants include heparin. Direct thrombin inhibitors and factor Xa inhibitors are new anticoagulants that inhibit platelet aggregation and thrombin activity [81]. Figure 1 illustrates that rhTM, ART-123, fibroblast growth factor 2 (FGF-2), and Annexin A5 all possess anticoagulant properties.

C1-esterase inhibitor (C1-INH) is a serine protease inhibitor that inhibits complement activation and reduces coagulation in the blood. In a clinical study involving 61 patients with sepsis, high doses of C1-INH have been shown to reduce inflammatory responses in patients with sepsis and significantly decrease the 28-day mortality rate to below 12% [8].

Antithrombin (AT) has anticoagulant and anti-inflammatory effects. Recombinant antithrombin gamma (RAT-γ) is a recombinant form of AT lacking fucose. A retrospective study of 49 patients with sepsis with antithrombin deficiency and DIC found that the RAT-γ (36 IU/kg) and AT treatment groups showed a significant reduction in Japanese Association for Acute Medicine-DIC and SOFA scores in patients with sepsis. Additionally, the coagulation disorders and organ failure in these patients were improved [9].

Recombinant hu-man soluble thrombomodulin (rhTM) can increase the expression of Glypcan1 and reduce the levels of inflammatory factors in the serum, improve cardiac microcirculation disorder and vascular injury caused by sepsis, and enhance the 48 h survival rate of lipopolysaccharide (LPS)-induced sepsis mouse models. A retrospective study on patients with DIC revealed that rhTM treatment significantly reduced the DIC score, C-reactive protein levels, and SOFA score, improving coagulation function, inflammatory response, and organ damage [10,11].

Recombinant human soluble thrombomodulin (ART-123) is a recombinant soluble form of human thrombomodulin. The multinational SCARLET phase III study included 800 patients and reported that ART-123 effectively improved abnormalities in patients with sepsis-associated coagulation disease, but did not significantly improve the 28-day survival rate [12].

FGF-2 improves blood coagulation and reduces the expression of TF and PAI1 in sepsis by inhibiting the protein kinase B (AKT)/mammalian target of rapamycin (mTOR)/p70 ribosomal protein subunit 6 kinase 1 (S6K1) phosphorylation. Moreover, it reduces the expression levels of plasminogen activator inhibitor-1 and tissue factor, thereby improving blood coagulation and alleviating injury to the lungs, liver, and other organs [13].

Annexin A5 can inhibit mitogen-activated protein kinase (MAPK) activation and reduces inflammation by blocking the binding of LPS and high-mobility group box protein 1 (HMGB1) to the Toll-like receptor 4 (TLR4)/myeloid differentiation protein-2 complex. A tail bleeding test in septic mice showed that annexin improved abnormal coagulation and ameliorated liver and kidney tissue damage caused by LPS and HMGB1 [14]. 

### 2.2. Antibacterial

Antimicrobial peptides have various biological activities, such as antibacterial, antiviral, antifungal, and immunomodulatory, and have first been identified in the pupae of the Saturn moth [82,83]. As shown in Figure 2, the primary antimicrobial peptides used for the prevention and treatment of sepsis are PS1-2, WIKE-14, and Pep19-2.5.

Polymyxin B (PMB) exerts both antibacterial and neutralizing functions. A prospective cohort study involving 60 patients with abdominal infectious sepsis found that polymyxin B-hemoperfusion (PMX-HP) treatment effectively reduced the level of endotoxin activity in patients with septic shock, but not improved their SOFA scores and survival rates [15]. Additionally, a meta-analysis of 12,234 critically ill adult patients showed that PMX-HP has a higher probability of causing nephrotoxicity in patients [16].

The PS1-2 peptide has shown enhanced antifungal activity through membrane dissolution and inhibited TLR2 activity and TNF-α expression levels in RAW264.7 cells. It can effectively combat sexually transmitted infections and inflammatory responses without triggering drug resistance [18].

KDEONWK-11 is a broad-spectrum antimicrobial peptide that reduces the immune response in patients with sepsis by neutralizing LPS, which involves the inhibition of biofilm metabolic activity and membrane formation [19].

WIKE-14, an antimicrobial peptide with antibacterial and anti-inflammatory activities, can neutralize endotoxins by inhibiting the binding of endotoxins to macrophages, thereby inhibiting the expression of inflammatory factors, and reduce reactive oxygen species (ROS) and nitric oxide (NO) production induced by LPS. In the LPS-induced sepsis mouse model, WIKE-14 effectively improved LPS-induced acute lung injury (ALI) [20].

Aspidasept (Pep19-2.5) is an anti-lipopolysaccharide polypeptide with a high binding affinity to LPS, neutralizes endotoxins, and exerts anti-inflammatory effects. In vitro cell and animal model toxicity tests demonstrate its safety as the therapeutic dose (30–50 μg/kg) is significantly lower than the toxic dose (2 mg/kg). Pep19-2.5 can reduce LPS-induced pyroptosis by preventing LPS from binding to TLR4, inhibiting the activation of nuclear factor kappa-light-chain-enhancer of activated B cells (NF-Κb) pathway, reducing the inflammatory response caused by LPS, and inhibiting the activation of caspase-1 [21,22].

### 2.3. Antioxidation

Antioxidants reduce oxidative damage by scavenging free radicals and inhibiting oxidation reactions [84]. As shown in Figure 3, α-chymotrypsin (α-ch), FGF-19, growth differentiation factor 7 (GDF7), milk fat globule epidermal growth factor 8 (MFG-E8), and mitsugumin53 (MG53) exert antioxidant effects in the prevention and treatment of sepsis.

Alpha-Chymotrypsin (α-ch) is a serine protease. By inhibiting the TLR4/NF-κB pathway, α-ch, a serine protease, reduces the levels of inflammatory cytokines IL-1β, IL-6, and TNF-α, and inhibits the expression levels of myeloperoxidase (MPO) and inducible nitric oxide synthase (iNOS) in a cecum ligation and puncture (CLP)-induced sepsis mouse model. After α-ch treatment, the survival rate of CLP mice increased by 50% [25].

FGF-19 can reduce the expression of linoleic acid (LA) and gamma linolenic acid, activate the nuclear factor erythroid 2-related factor 2/heme oxygenase-1 (HO-1) pathway, and inhibit the expression of iNOS, ROS, caspase-3, and cytochrome c, thereby alleviating lipid metabolism disorders, liver injury, and kidney injury in mice with LPS-induced sepsis [26]. 

GDF7 can reduce inflammation and oxidative stress by downregulating stimulator of interferon genes, promoting adenosine 5′monophosphate (AMP)-activated protein kinase (AMPK) phosphorylation, and improving LPS induced lung injury in septic mice; the 72 h survival rate of septic mice treated with GDF7 exceeds 50% [27].

As a biomarker of sepsis, MFG-E8 significantly reduces the levels of inflammatory factors and the concentration of malondialdehyde (MDA) and ferrous ions in the liver, increases the ratio of glutathione (GSH) and glutathione peroxidase 4 in the antioxidant system, and inhibits oxidative stress and ferroptosis in the liver of the CLP mouse model. MFG-E8 has a protective effect on liver injury and increases the survival rate of CLP mice by >40% [28].

MG53 is a protein of the tripartite motif family that exhibits anti-inflammatory and anti-oxidative properties, promotes plasma membrane repair, and improves myocardial dysfunction caused by sepsis by upregulating peroxisome proliferator activated receptor-alpha expression. Additionally, treatment with MG53 has been shown to increase the 72 h survival rate (>60%) of CLP rats [29].

### 2.4. Anti-Apoptosis

Anti-apoptosis drugs inhibit apoptosis by regulating the ratio of anti-apoptotic proteins (Bcl-2 and Mcl1) to pro-apoptotic proteins (Bax and Bad) as well as inflammation and oxidative stress [85]. As shown in Figure 4, hepatocyte growth factor (HGF), IL-7, IL-15, IL-22, Ac2-26, Vaspin, Hsp22, and adiponectin (APN) exhibit anti-apoptotic functions in the prevention and treatment of sepsis.

HGF inhibits phosphoinositide 3-kinase (PI3K) and AKT phosphorylation, activates the mTOR pathway, and inhibits the activation of NF-κB through target cellular mesenchymal–epithelial transition factor; reduces apoptosis in the liver and lung tissue; reduces the IL-1 β, IL-18, lactated hydrogenase (LDH), and ROS levels; and alleviates sepsis-induced inflammatory and oxidative stress. HGF treatment has been shown to improve the survival rate of CLP mice to over 50% [30]. IL-7 promotes the secretion of B-cell lymphoma-2 (Bcl-2) and interferon-γ (IFN-γ) through an activated kinase (JAK), promotes signal transducer and activator of transcription (STAT) phosphorylation, and inhibits the expression of pro-apoptotic protein Bcl-like 11 (Bim).A preclinical trial including 70 patients with septic shock demonstrated that IL-7 restored the CD4+ and CD8+ T cell levels in patients with sepsis and improved the lymphocyte dysfunction caused by sepsis [31]. IL-15, a cytokine belonging to the IL-2 family, can increase the ratio of T cells and natural killer cells in septic rats by promoting the expression of Bcl-2 and IFN- γ and inhibiting the expression of pro-apoptotic protein, Bim. After IL-15 injection, the survival rates of CLP mice and mice infected with *Pseudomonas aeruginosa* increased by three- and two-fold, respectively [32].

IL-22, a member of the IL-10 family of cytokines, can induce the differentiation and development of inhibitory M2 macrophages by promoting the expression of S100 calcium-binding protein A9 (S100A9) in the liver, reducing inflammation, and activating STAT3 and transcription factor 4, which are recombinant autophagy related protein 7 hepatocyte autophagy signaling pathways to reduce liver inflammation and injury [33,86]. 

IL-38 is a cytokine of the IL-1 family that regulates the NOD-like receptor thermal-protein-domain-associated protein 3 (NLRP3)/IL-1β signaling pathway, inhibits the activation of NLRP3 inflammasomes, enhances CD4 CD25 Treg cell immune function, reduces the expression of inflammatory mediators and pro-apoptotic proteins, attenuates early inflammatory response and apoptosis, improves the depletion of effector T cells during sepsis, and increases the survival rate of septic mice (>60%) [34,35]. FGF-15 is an analog of FGF-19, which significantly improves liver inflammation and apoptosis in septic mice by targeting fibroblast growth factor receptor 4, reducing the proportion of Tregs, and increasing the survival rate of septic mice by 40% [36]. Insulin-like growth factor 1, which is an anti-apoptosis factor, reduces hippocampal apoptosis and improves cognitive impairment by inhibiting the overexpression of cytochrome c and tumor necrosis factor receptors in a rat model of septic encephalopathy (SE) [38]. Maf1 is a transcriptional regulator of RNA polymerase III that reduces inflammation and apoptosis by competitively binding to the NLRP3 and inhibiting the expression and activity of NLRP3. Maf1 could also be used to prevent sepsis-associated encephalopathy (SAE) [39]. Ac2-26 is an N-terminal active peptide of Annexin A1 that targets formyl peptide receptor-2 receptors and inhibits PI3K and AKT phosphorylation. Ac2-26 downregulates the level of NF-κB, reduces the expression of inflammatory cytokines and pro-apoptotic proteins in septic mice, inhibits inflammation and apoptosis, improves renal injury, and increases the 7-day survival rate of CLP mice (>40%) [40].

Vaspin (Serpin A12) is an adipose factor of the Serpin family that inhibits the expression of kallikrein 7; downregulates the expression of pro-apoptotic protein Bcl2-associated X (bax), TNF-α, and other inflammatory factors; promotes the expression of anti-apoptotic protein bcl-2; reduces inflammatory response and apoptosis; and improves septic heart injury in mice. Vaspin treatment increased the survival rate of CLP mice by 50% [41].

Heat shock protein 22 (Hsp22/HspB8) functions as a molecular chaperone and an anti-inflammatory and anti-oxidation molecule. Hsp22 may inhibit the expression of apoptotic proteins and inflammatory factors by promoting AMPK activation and reducing mTOR phosphorylation levels, and has been shown to significantly improve myocardial injury and inflammation in an LPS-induced sepsis mouse model [37].

APN, secreted by the adipocytes, activates the PI3K/AKT pathway, reduces the proportion of the apoptotic proteins bax and cleaved-caspase-3 induced by LPS, and improves cardiomyocyte apoptosis induced by sepsis. In addition, APN inhibits mTOR activation by promoting AMPK phosphorylation, regulating the AMPK/mTOR pathway, and reducing inflammatory responses and liver injury in septic rats [42,43].

Irisin is a polypeptide hormone produced by the cleavage of membrane protein fibronectin type III domain-containing protein 5. Irisin can decrease the IL-1 β and Gasdermin D (GSDMD) levels by regulating the mitochondrial ubiquitin ligase (MITOL)/GSDMD pathway, improving the levels of LDH and creatine kinase-MB, and reducing inflammation and cardiomyocyte scorch caused by sepsis [44].

### 2.5. Regulators of Blood Pressure and Volume 

Patients with sepsis often have low blood pressure and insufficient blood volume; therefore, they must be regulated via in vitro injection of drugs. HSA, selepressin, and B38-CAP improve blood volume and hypotension during sepsis treatment.

HSA regulates the blood osmotic pressure [87] and improves edema, hypoalbuminemia, glycocalyx degradation and hypovolemia caused by sepsis. A retrospective study involving 5009 patients with sepsis demonstrated that treatment with 5% HSA significantly reduced sepsis-related 28-day mortality (8.7%) [45]. In addition, albumin can also reduce heme-induced hepatic sinusoidal contraction and cytotoxicity, and improve liver microcirculation and organ function by binding heme. Albumin in mouse models of pulmonary ischemia-reperfusion reduces the risk of pulmonary edema and improves endothelial dysfunction [88,89].

Selepressin is a selective vasopressin (VP) V1a agonist. A phase IIa randomized, placebo-controlled trial with 53 patients with septic shock showed that selepressin (2.5 ng/kg/min) could replace norepinephrine (18.24 μg/kg/min) to maintain the mean arterial pressure balance with good tolerance [46]. However, a phase III/IIb clinical study with 868 patients with septic shock revealed that selepressin did not significantly reduce the usage of ventilators and pressor drugs in patients with septic shock within 30 days [47].

B38-CAP, an ACE2-like enzyme derived from Bacillus-like bacteria, can upregulate the level of angiotensin 1–7, reduce the levels of Ang II and inflammatory cytokines and chemokines, improve lung injury caused by sepsis, and increase the survival rate of CLP mice by 20% [48].

VP can increase the mean arterial pressure, improve body perfusion, and reduce the number of renal replacement therapy in patients with septic shock in clinical practice, but there is no significant improvement in patient mortality and ICU length of stay [79,80].

### 2.6. Regulators of Inflammation

Some anti-inflammatory cytokines and cytokine inhibitors or antibodies, as well as anti-inflammatory proteins and peptides, have demonstrated anti-inflammatory activities [90]. Figure 5 illustrates the anti-inflammatory mechanisms of S100A8, crotoxin (CTX), Apelin-13, and secretory leukocyte protease inhibitor (SLPI).

Coenzyme Q10 significantly increases the level of Beclin1, decreases the levels of liver injury indices (alanine aminotransferase, alkaline phosphatase, and aspartate aminotransferase) and inflammatory mediators (IL-6, IL-1 β, TNF-α, and NLRP3) in CLP mice, and reduces liver inflammation and injury [49].

Erythropoietin (EPO) is a glycoprotein with a molecular weight of 30 kDa. A retrospective study involving 344 patients with sepsis demonstrated that high-dose EPO (8000–16,000 units/week) improved the 29-day survival rate (85%), without increasing the incidence of adverse events, such as thrombosis. Furthermore, high-dose EPO reduced mortality and inflammation in a septic mouse model [50].

S100 calcium-binding protein A8 (S100A8) is a calcium-binding protein that belongs to the S100 family. S100A8 can reduce the expression of inflammatory cytokines and chemokines by acting on TLR4 and can directly bind to inflammatory cytokines IL-6 and TNF-α to reduce the inflammatory response and prolong the survival time of septic mice [51].

CTX is a protein derived from South American rattlesnake venom. In the model of CLP sepsis, CTX significantly downregulated the expression of pro-inflammatory cytokines TNF-α and IL-6, and increased the ratio of lipoxin A4, prostaglandin E2, and IFN-γ in mice plasma. In the bone-marrow-derived macrophage of mice with sepsis, CTX could improve the inflammatory response caused by sepsis and reduce the mortality of CLP mice by regulating phagocytosis and promoting the secretion of ROS and NO. Moreover, it reduced the mortality rates of CLP mice by 40% [52].

Human chorionic gonadotropin, a hormone first identified in the placenta, can reduce the expression of pro-inflammatory mediators (TNF-α, IL-6, and Pentraxin 3) and chemokines (macrophage inflammatory protein 1-a and chemotokine ligand 5) caused by sepsis, and improve organ damage caused by sepsis [53].

Granulocyte-macrophage colony-stimulating factor (GM-CSF) is a glycoprotein. In patients with liver cirrhosis and sepsis, GM-CSF therapy regulates the ratio of myeloid-derived suppressor cells and Tregs and improves immune function of CD4+ T cells [55]. β15-42 is a fibrin-derived peptide that can improve the barrier function of capillaries, reduce capillary leakage, inhibit the transfer of inflammatory cells, reduce the expression of inflammatory factors in the liver and lung tissues, reduce the infiltration of inflammatory cells into the liver and lung tissues, and ameliorate the inflammation and injury of the liver and lung tissues caused by sepsis [54]. Apelin is an APJ receptor ligand. Apelin-13 downregulates the production of NADPH oxidase 4 and ROS through the APJ receptor, inhibits macrophage glycolysis, and reduces pulmonary inflammation in mice with sepsis. In addition, Apelin-13 can improves inflammatory cell infiltration and reduce pulmonary fibrosis caused by sepsis by reducing the expression of transforming growth factor beta and drosophila mothers against decapentaplegic protein (SMAD) 2, and SMAD 3 [56,57].

SLPI is a kind of whey acid family protein, which can effectively reduce the level of systemic inflammatory response and reduce the expression of cytokines, such as IL-6, TNF-α, IL-1, and monocyte chemoattractant protein-1 by inhibiting the NF-κB and MAPK pathways [17].

AB103 is a CD28 antagonist that inhibits inflammation and improves the efficiency of bacterial clearance in a bacterial sepsis model. A multicenter, randomized clinical trial involving 40 patients with necrotizing soft-tissue infections showed that AB103 could improve organ damage and safety. However, it did not significantly affect the duration of ventilator use or the level of cytokines [23,24]. 

## 3. Antibody Drugs

Antibody drugs are preparations composed of antibodies with the advantages of high specificity and long half-life. According to their structure, antibody drugs can be divided into immunoglobulin G (IgG), antibody conjugates, and antibody fragments, and have been widely used to treat tumors, inflammation, and immunity [91]. As shown in Figure 6, antibody drugs currently used in clinical trials or laboratory studies on sepsis include anti-HMGB1 antibodies, atezolizumab, adrecizumab, and tocilizumab.

IVIG (intravenous immunoglobulins) is a mixed IgG antibody. A single-center retrospective study of 239 patients with sepsis showed that IVIG significantly improved the prognosis of patients with low IgG levels and reduced mortality. A retrospective multicenter study of 850 patients with severe COVID-19 showed that IVIG did not significantly improve the 28-day survival rate of patients, and might have induced adverse thrombosis and allergic reactions [58,59].

HMGB1 is an inflammatory mediator involved in sepsis. Anti-HMGB1 antibody is a monoclonal antibody developed against the inflammatory mediator HMGB1 that can regulate the levels of pro-inflammatory and anti-inflammatory factors, improve the inflammatory response in CLP mice, restore hemoglobin and hematocrit, improve anemia, enhance anti-infection ability, and reduce the mortality rate of CLP mice by 50% [60,61].

Atezolizumab is a programmed cell death-ligand 1 (PD-L1) monoclonal antibody, which can treat sepsis by blocking the programmed cell death protein 1 (PD-1)/PD-L1 pathway. In sepsis, atezolizumab can reduce the level of endotoxins in the CLP model, increase the expression levels of claudin-1 and occludin proteins, improve intestinal barrier function, reduce T cell apoptosis, and alleviate immunosuppression, and reduce the mortality of CLP mice by 25% within 90 h of treatment [62].

Vilobelimab is a monoclonal antibody against the allergic toxin complement component 5a (C5a) that ameliorates infectious organ dysfunction by neutralizing C5a. A randomized, double-blind, multicenter phase IIa study of 72 patients with sepsis or septic shock showed that vilobelimab neutralizes C5a in a dose-dependent manner with good tolerance and safety. However, the study did not show significant improvement in sepsis prognosis and mortality [63]. 

Secukinumab is a human monoclonal antibody that neutralizes IL-17A to inhibit the inhibitor kappa B alpha (IκBα) /NF-κB pathway, reduce IκBα and NF-κB phosphorylation levels, inhibit the expression of pro-inflammatory factors (i.e., IL-6 and TNF-α), and improve the inflammatory response and lung injury induced by sepsis. Simultaneously, the survival rate of CLP rats injected with a high dose of secukinumab (20 mg/kg) was 60% after 196 h [64].

Adrecizumab (HAM8101) is a non-neutralizing human adrenomedullin monoclonal antibody that reduces the expression of vascular endothelial growth factor, increases the level of angiopoietin-1, improves the effect of sepsis on vascular permeability, and reduces inflammation and mortality in septic mice [65]. A phase II, double-blind, randomized, controlled trial involving 301 patients with septic shock found that adrecizumab was well tolerated but did not significantly improve the survival rate of patients [92].

Tocilizumab is an IL-6 receptor antibody that can inhibit the expression of S100A12 and NLRP3, reduce the levels of pro-inflammatory factors, such as IL-6 and TNF-α; regulate the levels of superoxide dismutase (SOD) and MDA; reduce inflammation, oxidative stress, and apoptosis in CLP mice; and improve acute lung injury caused by sepsis [66].

## 4. Nucleic Acid

MicroRNA drugs are single-stranded miRNA drug preparations containing 19–24 nucleotides that do not encode proteins and act mainly at the post-transcriptional stage. MicroRNA drugs have been used in the treatment of cancers and other diseases [93]. As shown in Figure 7, the miRNA drugs currently used in sepsis research include miR-25-5p, miR-214-3p, miR-142-5p, miR-340-5p, miR-26a5p, and miR-490-3p.

miR-25-5p can improve brain injury caused by sepsis by inhibiting the thioredoxin-interacting protein TXNIP/NLRP3 pathway; reducing the levels of TXNIP, NLRP3, and cleaved caspase-1; and inhibiting the expression of inflammatory factors (such as IL-6, IL-1β, and TNF-α and peroxide). Moreover, it can reduce LPS-induced inflammation, oxidative stress, and apoptosis, and improve LPS-induced brain injury in septic rats [67]. miR-214-3p can inhibit the expression of cathepsin B, regulate the levels of anti-apoptotic bcl-2 and pro-apoptotic proteins Bax and caspase-3, decrease the levels of ROS and MDA, increase the proportion of SOD, reduce the apoptosis and oxidative damage of AC16 cells stimulated by LPS, and improve myocardial injury caused by sepsis [68]. miR-340-5p can reduce the expression of myeloid differentiation factor 88 protein, inhibit the elevated production of ROS and MDA induced by LPS, increase the level of GSH, reduce the oxidative stress response of sepsis-induced cardiomyopathy (SIC) cell model, and improve LPS-induced HL-1 cell injury. miR-340-5p can reduce the degree of myocardial injury and oxidative stress in LPS sepsis-induced cardiomyopathy model [69]. miR-26a-5p can inhibit the expression of inflammatory factors (TNF-α, IL-1β, and IL-6) and pro-apoptotic protein Bax, reduce pulmonary inflammation and apoptosis in mice, improve pulmonary inflammation and apoptosis, and increase the survival rate of septic ALI mice by 30% [70]. By reducing the expression levels of IL-1 receptor-associated kinase 1 (IRAK1), miR-490-3p inhibits the IRAK1/ TNF receptor-associated factor 6 (TRAF6) pathway; reduces the level of NF-κB phosphorylation; decreases the expression of inflammatory factors IL-1β, IL-6, and TNF α in ALI rat model; and improves LPS-induced pulmonary inflammation and apoptosis in rats [71]. Furthermore, the inhibition of the NF-κB pathway through the targeting of IRAK1/TRAF6 by miR-146a has been observed to effectively reduce the inflammatory response in mice subjected to CLP, as well as mitigate splenocyte apoptosis. Notably, the administration of miR-146a resulted in a significant improvement in sepsis-induced organ damage, leading to a noteworthy 40% increase in the survival rate of CLP mice (*n* = 15) [72].

## 5. Polysaccharides and Other Macromolecules

Polysaccharides are macromolecular compounds composed of monosaccharides connected by glycosidic bonds and are used in many fields, such as anti-tumor, anti-inflammation, antioxidation, and immune regulation. 

Polysaccharides have the advantages of fewer side effects and good safety [94]. As shown in Figure 8, the polysaccharides macromolecules currently used for the treatment of sepsis are unfractionated heparin (UFH), N-acetylheparin (NAH), lentinan, *Lycium barbarum* polysaccharides (LBPs), and *Poria cocos* polysaccharide (PCP). In addition, polyphenol tannic acid and the lipid derivative eritoran are used in the prevention and treatment of sepsis.

UFH, a glycosaminoglycan, and NAH, a non-anticoagulant, can reduce lung leukocyte infiltration, capillary barrier injury, and pulmonary edema in septic mice; inhibit caspase-11-induced cell pyrolysis induced by LPS and HMGB1; reduce the increase in pro-inflammatory cytokines IL-1 β and TNF-α; and inhibit heparinase activity to protect endothelial glycocalyx. The anti-inflammatory effect of UFH is superior to that of NAH; however, UFH increases the risk of bleeding. A retrospective study of 3377 patients with SIC found that UFH (6250–13,750 IU/d) reduced the risk of death in patients with a SIC score of 4 points; however, UFH increased the risk of bleeding in clinical trials [73,95,96].

Lentinan is a polysaccharide isolated from *Lentinus edodes* that has anti-inflammatory, antitumor, and immune functions. Lentinan can reduce the expression of pro-inflammatory factors, such as TNF-α, IL-1 β, IL-6, and HMGB1; improve intestinal inflammation and oxidative stress induced by sepsis; regulate the expression of bcl-2 and bax by inhibiting NF-κB pathway; and inhibit apoptosis in order to improve intestinal injury [74].

LBP is the primary bioactive component of the traditional Chinese medicine Lycium barbarum and exerts anti-inflammation and antioxidation effects. LBP promotes pyruvate kinase muscle isoform 2 (PKM2) ubiquitination by increasing the expression of NEDD4-like E3 ubiquitin protein ligase (Nedd4L), neural precursor cell expressed developmentally downregulated protein 4 (Nedd4), and G-protein subunit β2 in Raw264.7 cells; reduces the level of PKM2; and inhibits LPS-induced glycolysis process and cell differentiation of M1 macrophages. LBP also reduces the expression of IL-1β, TNF-α, and HMGB1 and alleviates the inflammatory response induced by LPS [75].

PCPs is a polysaccharide component of the traditional Chinese medicine *Poria cocos*, with various biological activities, such as anti-inflammation, antioxidation, and immune regulation. In a septic mice model, PCP preconditioning significantly decreased the expression of inflammatory cytokines IL-6, IL-1 β, and TNF-α, improved inflammation and oxidative stress in FP mice by downregulating the levels of MDA and MPO. PCP can improve the inflammation and oxidative stress response of fecal-induced peritonitis mice, reduce splenocyte apoptosis, regulate the proportion of spleen Tregs, and increase the survival rate of sepsis mouse models by approximately 30% [76].

Tannic acid is a polyphenolic compound with both anti-inflammatory and anxiolytic properties. In sepsis, tannic acid can regulate blood pressure; reduce the expression of TNF-α, IL-6, IL-1 β and MDA in the brain tissue; upregulate the expression of γ-aminobutyric acid sub-type A receptors in the hippocampus; reduce inflammation and oxidative stress in patients with sepsis; decrease inflammation and oxidative stress in CLP rats; and improve anxiety-related behaviors in SAE rats [77].

Eritoran is a derivative ofLPS lipid A and a TLR4 blocker. Eritoran can inhibit the expression of inflammatory mediators, inhibit the cytokine storm in sepsis, increase Th1 reaction time, and reduce death in mice with bacterial sepsis. However, in a randomized, double-blind, phase III clinical trial of 1961 participants, eritoran could not reduce the 28-day mortality in patients with severe sepsis [78,97]. 

## 6. Conclusions

The efficacy, dosage, side effects, and therapeutic mechanisms of biological macromolecules remain unclear and only a few have entered the clinical stage. Moreover, clinical data are limited, and larger clinical trials are needed to clarify these. Peptides and proteins are the most widely studied biological macromolecules for the prevention and treatment of sepsis, some of which have been newly discovered and synthesized, and most are in the preliminary research stage. At present, peptides and proteins in the clinical treatment of sepsis are mainly used for anticoagulation and regulation of blood pressure. The antibodies used in sepsis belong to the “new use of old drugs”, mainly aimed at the inflammatory reaction caused by sepsis. Only a few drugs, such as IVIG and vilobelimab, have entered the clinical stage. Nucleic acid drugs are microRNAs designed to inhibit the expression of proteins related to inflammation and apoptosis pathways and are currently in the preliminary research stage. Polysaccharide biomolecules have anticoagulant, anti-inflammatory, anti-apoptotic, and other biological functions; however, only heparin is used in the clinical treatment of sepsis.

Compared with small molecules, biomacromolecules have advantages such as precise targeting, long half-life, low toxicity, and fewer side effects. Their large spatial structure enables more stable target binding. However, biomacromolecules also have several shortcomings. In comparison with small molecules, the preparation process of biomacromolecules is immature. Most biomacromolecules, such as protein and nucleic acid drugs, require low-temperature storage to prevent degradation and have higher requirements for storage and transportation equipment. Biomacromolecules, owing to their larger molecular weight, are difficult to absorb and are prone to immunogenicity. Additionally, the biological activity of biomacromolecules such as proteins, polypeptides, and nucleic acids relies on their spatial structure, making oral administration challenging. Therefore, optimizing the research and development process of molecular preparation to reduce production costs and improve product purity is essential. Currently, there are relatively few immunogenicity challenges, such as the humanization of antibodies. Overcoming the challenges of absorption can be addressed by identifying therapeutic fragments of proteins and peptides, reducing molecular weight, and enhancing absorption efficiency through nanoparticle conjugation or cooperation with cell-penetrating peptides. Coating methods and other approaches can also facilitate the oral administration of macromolecules [6,7,98].

Macromolecular drugs demonstrate superior therapeutic efficacy, tolerability, and safety. The therapeutic dose of certain macromolecules is lower than that of small molecules, and macromolecules have a longer half-life, which enhances the therapeutic effect. Resistance to treatment with antimicrobial peptide macromolecules has also been reported. Furthermore, the therapeutic dose of macromolecular drugs is significantly lower than their toxic dose, indicating a higher level of safety. Therefore, we believe that macromolecular drugs hold great promise in the treatment of sepsis.

## Figures and Tables

**Figure 1 ijms-24-13017-f001:**
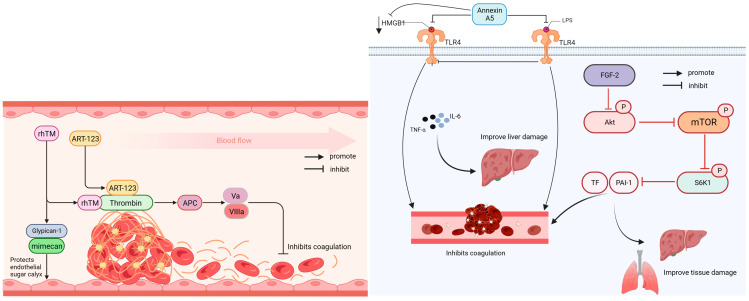
Anticoagulation mechanism of rhTM, ART-123, Annexin, and FGF-2. AKT: protein kinase B; APC: activated protein C; ART-123: Recombinant hu-man soluble thrombomodulin; FGF-2: fibroblast growth factor 2; HMGB1: high-mobility group box protein 1; IL-6: interleukin-6; LPS: lipopolysaccharide; mTOR: mammalian target of rapamycin; PAI-1: plasminogen activator inhibitor-1; rhTM: Recombinant human soluble thrombomodulin; S6K1: p70 ribosomal protein subunit 6 kinase 1; TF: tissue factor; TLR4: Toll-like receptor 4; TNF-α: tumor necrosis factor; VIIIa: activated factor seven; Va: activated factor five.

**Figure 2 ijms-24-13017-f002:**
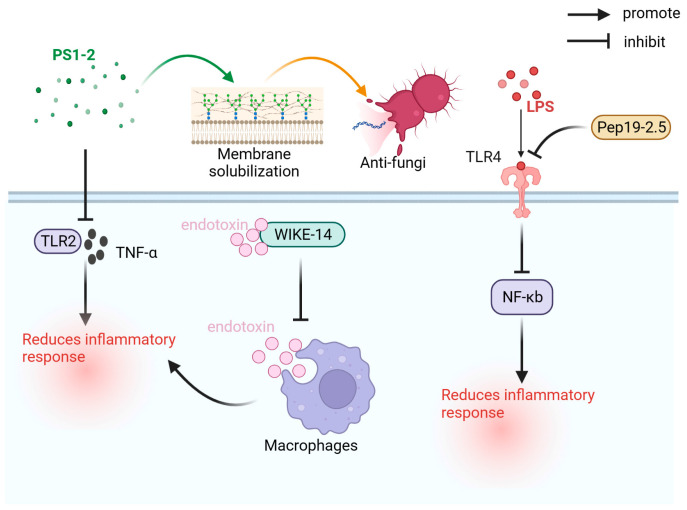
Antibacterial mechanism of WIKE-14, PS1-2, and Pep19-2.5. LPS: lipopolysaccharide; NF-κB: nuclear factor kappa-light-chain-enhancer of activated B cells; Pep19-2.5: aspidasept; TLR2: Toll-like receptor 2; TLR4: Toll-like receptor 4; TNF-α: tumor necrosis factor.

**Figure 3 ijms-24-13017-f003:**
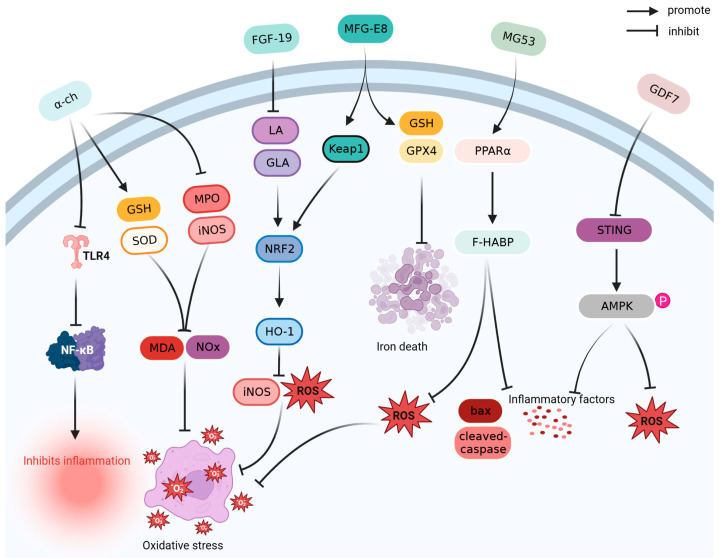
Antioxidant mechanism of α-ch, FGF-19, MFG-E8, MG53, and GDF7. AMPK: adenosine 5′-monophosphate (AMP)-activated protein kinase; Bax: BCL2-associated X; FGF-19: fibroblast growth factor 19; GDF7: growth differentiation factor 7; GLA: gamma linolenic acid; GPX4: glutathione peroxidase 4; GSH: glutathione; HO-1: heme oxygenase-1; iNOS: inducible nitric oxide synthase; Keap1: Kelch-like ECH-associated protein 1; LA: linoleic acid; MDA: malondialdehyde; MFG-E8: milk fat globule epidermal growth factor 8; MG53: mitsugumin-53; MPO: myeloperoxidase; NF-κB: nuclear factor kappa-light-chain-enhancer of activated B cells; NOx: nitrite/nitrate; NRF2: nuclear factor erythroid 2-related factor 2; PPARα: peroxisome proliferator-activated receptor-alpha; ROS: reactive oxygen species; SOD: superoxide dismutase; STING: stimulator of interferon genes; TLR4: Toll-like receptor 4; α-ch: alpha-chymotrypsin.

**Figure 4 ijms-24-13017-f004:**
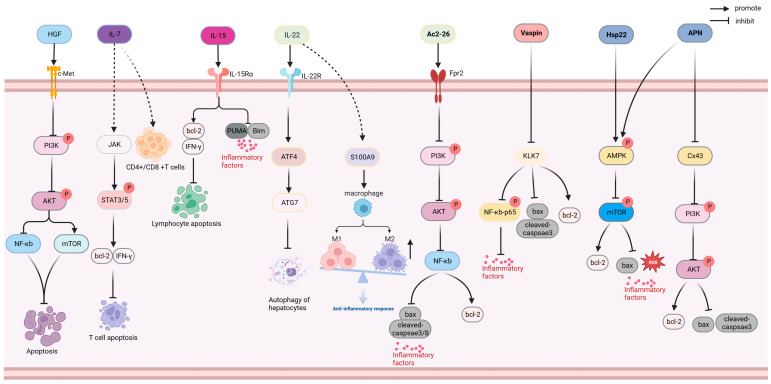
Anti-apoptotic mechanism of HGF, IL-7, IL-15, IL-22, Ac2-26, Vaspin, Hsp22, and APN. AKT: protein kinase B; AMPK: adenosine 5′-monophosphate (AMP)-activated protein kinase; APN: adiponectin; ATF4: activating transcription factor 4; ATG7: autophagy related protein 7; Bax: BCL2-associated X; bcl-2: B-cell lymphoma-2; Bim: Bcl-2-like protein 11; c-Met: cellular mesenchymal–epithelial transition factor; HGF: hepatocyte growth factor; Hsp22: heat shock protein 22; IFN-γ: interferon-γ; IL-15: interleukin-15; IL-22: interleukin-22; IL-7: interleukin-7; JAK: Janus Kinase; KLK7: kallikrein 7; mTOR: mammalian target of rapamycin; NF-κB: nuclear factor kappa-light-chain-enhancer of activated B cells; PI3K: phosphoinositide 3-Kinase; PUMA: p53 upregulated modulator of apoptosis; ROS: Reactive oxygen species; S100A9: S100 calcium-binding protein A9; STAT3/5: signal transducer and activator of transcription 3/5; Vaspin: Serpin A12.

**Figure 5 ijms-24-13017-f005:**
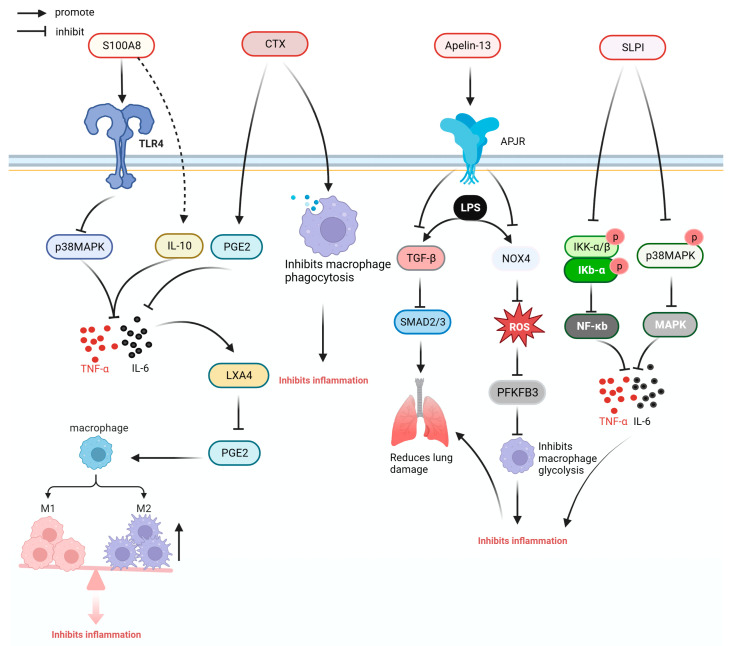
Anti-inflammatory mechanism of S100A8, CXT, Apelin-13, and SLPI. APJR: apelin receptor; CTX: crotoxin; IKK-α/β: inhibitor of kappa B kinase-α/β; IL-10: interleukin-10; IL-6: interleukin-6; Iκb-α: inhibitor kappa B alpha; LPS: lipopolysaccharide; LXA4: lipoxin A4; MAPK: mitogen-activated protein kinase; NF-κB: nuclear factor kappa-light-chain-enhancer of activated B cells; NOX4: NADPH oxidase 4; p38 MAPK: P38 mitogen-activated protein kinases; PFKFB3: 6-Phosphofructo-2-kinase/fructose-2,6-bisphosphatase; PGE2: prostaglandin E2; ROS: reactive oxygen species; S1000A8: S100 calcium-binding protein A8; SLPI: secretory leukocyte protease inhibitor; SMAD2/3: drosophila mothers against decapentaplegic protein 2/3; TGF-β: transforming growth factor beta; TLR4: Toll-like receptor 4; TNF-α: tumor necrosis factor.

**Figure 6 ijms-24-13017-f006:**
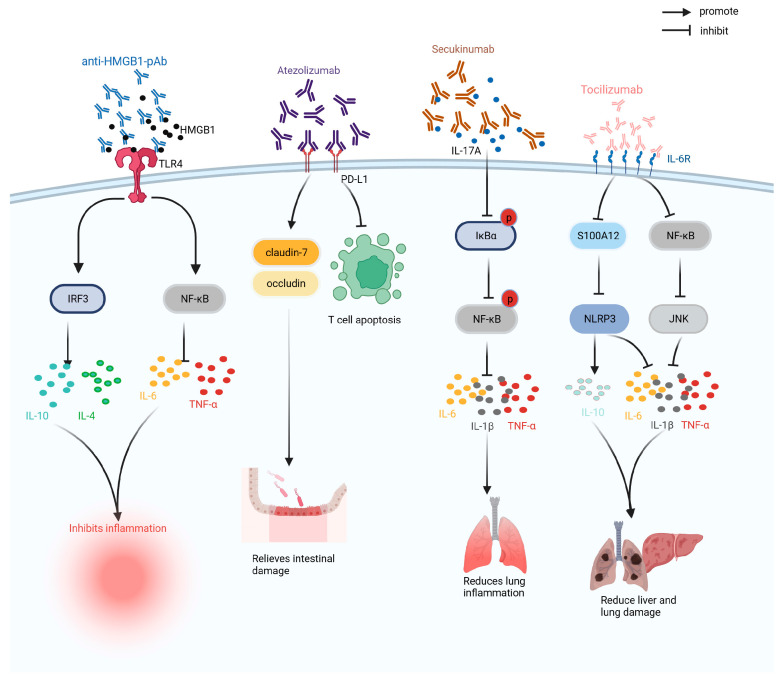
Therapeutic mechanism of anti-HMGB1pAb, atezolizumab, secukinumab, adrecizumab, and tocilizumab. HMGB1: high-mobility group box chromosomal protein 1; IL-10: interleukin-10; IL-1β: interleukin-1β; IL-4: interleukin-4; IL-6: interleukin-6; IRF3: interferon regulatory factor 3; IκBα: inhibitor kappa B alpha; JNK: Jun N-terminal Kinase; NF-κB: nuclear factor kappa-light-chain-enhancer of activated B cells; NLRP3: NOD-like receptor thermal-protein-domain-associated protein 3; PD-L1: programmed cell death-ligand 1; S100A12: S100 calcium-binding protein A12; TLR4: Toll-like receptor 4; TNF-α: tumor necrosis factor.

**Figure 7 ijms-24-13017-f007:**
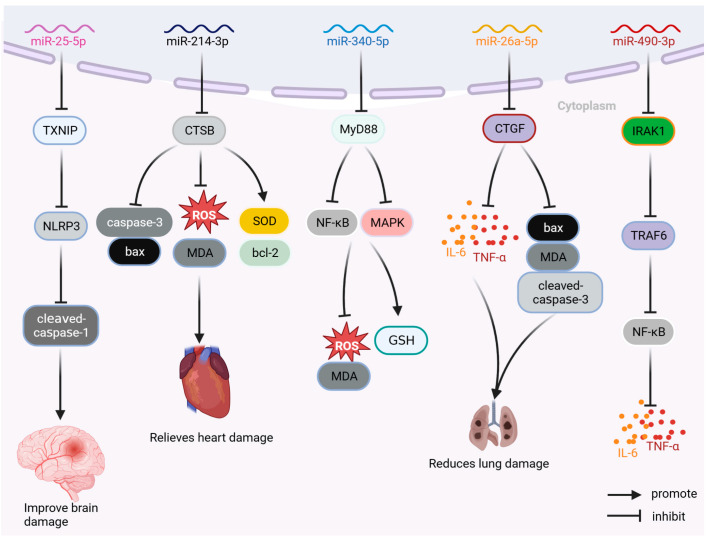
Therapeutic mechanism of miR-25-5p, miR-214-3p, miR-142-5p, miR-340-5p, miR-26a-5p, and miR-490-3p. Bax: BCL2-associated X; bcl-2: B-cell lymphoma-2; CTGF: connective tissue growth factor; CTSB: cathepsin B; GSH: glutathione; IL-6: interleukin-6; IRAK1: interleukin 1 receptor-associated kinase 1; MAPK: mitogen-activated protein kinase; MDA: malondialdehyde; miR: microRNA; MyD88: myeloid differentiation factor 88; NF-κB: nuclear factor kappa-light-chain-enhancer of activated B cells; NLRP3: NOD-like receptor thermal-protein-domain-associated protein 3; ROS: reactive oxygen species; SOD: superoxide dismutase; TNF-α: tumor necrosis factor; TRAF6: TNF receptor-associated factor 6; TXNIP: thioredoxin-interacting protein.

**Figure 8 ijms-24-13017-f008:**
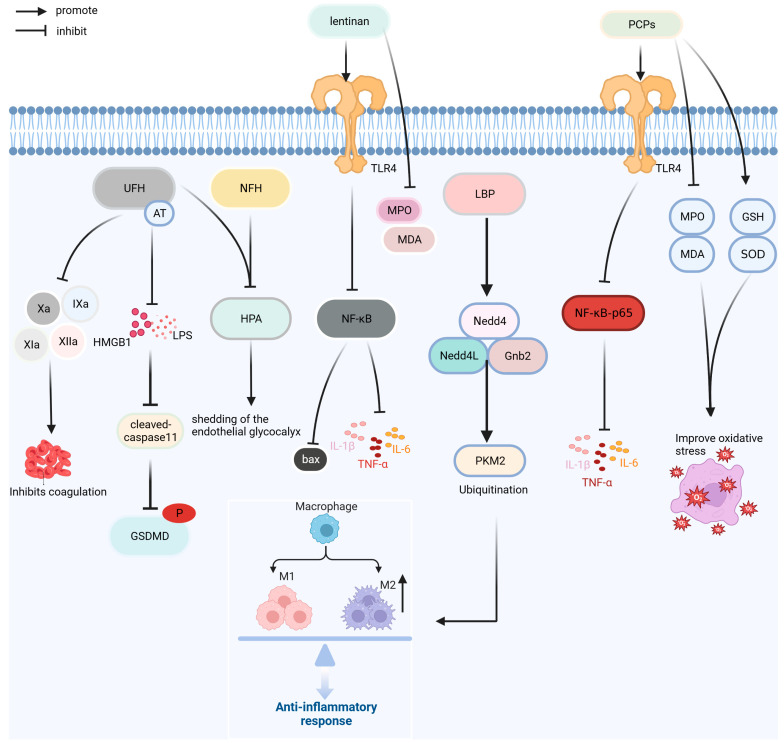
Treatment mechanism of UFH, NAH, lentinan, LBP, and PCP. Bax: BCL2-associated X; Gnb2: G protein subunit β2; GSDMD: Gasdermin D; GSH: glutathione; HMGB1: high-mobility group box protein 1; HPA: heparinase; IL-1β: interleukin-1β; IL-6: interleukin-6; LBP: Lycium barbarum polysaccharide; LPS: lipopolysaccharide; MDA: malondialdehyde; MPO: myeloperoxidase; NAH: N-acetyl heparin; Nedd4: neural precursor cell expressed developmentally downregulated protein 4; Nedd4L: NEDD4-like E3 ubiquitin protein ligase; NF-κB: nuclear factor kappa-light-chain-enhancer of activated B cells; PCPs:Poria cocos polysaccharides; PKM2: Pyruvate kinase muscle isoform 2; SOD: superoxide dismutase; TLR4: Toll-like receptor 4; TNF-α: tumor necrosis factor; UFH: unfractionated heparin; Xa: factor Xa.

**Table 1 ijms-24-13017-t001:** Summary of sepsis macromolecules.

Serial Number	Name	Target/Pathway	Mechanism of Action	Effect	Research Progress	Reference Number
1	C1-esterase inhibitor (C1-INH)		Inhibits complement and thrombin activation	Reduces mortality	Clinical trials	[8]
2	Recombinant antithrombin gamma (RAT-γ)		Inhibits coagulation factor	Improves patient outcomes	Clinical trials	[9]
3	Recombinant human soluble thrombomodulin (rhTM)	Thrombin	Improves coagulation function	Improves blood clotting	Clinical trials	[10,11]
4	Recombinant human soluble thrombomodulin (ART-123)	Thrombin	Reduces the ratio of D-dimer and TATc	Improves blood clotting	Clinical trials	[12]
5	Fibroblast growth factor 2 (FGF-2)	AKT/mTOR/S6K1 pathway	Suppresses the AKT/mTOR/S6K1 pathway	Improves blood clotting	Animal experiment	[13]
6	Annexin A5	TLR4	Inhibits LPS and HMGB1 binding of the TLR4	Improve blood clotting	Animal experiment	[14]
7	Polymyxin B		Neutralizes endotoxin		Clinical trials	[15,16]
8	Secretory leukocyte protease inhibitor	NF-κB and MAPK pathways	Inhibits the NF-κB and MAPK pathways	Inhibit inflammation	Animal experiment	[17]
9	PS1-2	NF-κB and AMPK pathways	Dissolves the membrane dissolution and inhibits the NF-κB and AMPK pathway	Inhibits fungal infections	Animal experiment	[18]
10	KDEON WK-11		Broad-spectrum antibacterial activity	Inhibits microbial activity	Cell experiment	[19]
11	WIKE-14		Neutralize endotoxin	Ameliorates LPS-induced acute lung injury	Animal experiment	[20]
12	Aspidasept (Pep19-2.5)	NF-κB pathway	Neutralizes endotoxin and inhibits the NF-κB pathway	Reduces pyroptosis	Animal experiment	[21,22]
13	AB103	CD28	CD28 antagonist	Ameliorates organ damage	Phase II clinical trial	[23,24]
14	Alpha-Chymotrypsin (α-ch)	TLR4/NF-κB pathway	Inhibits the TLR4/NF-κB pathway	Improves the survival rates	Animal experiment	[25]
15	Fibroblast growth factor 19 (FGF-19)	NRF2/HO-1 pathway	Activates the NRF2/HO-1 pathway	Ameliorates organ damage	Animal experiment	[26]
16	Growth Differentiation Factor 7 GDF-7	STING/AMPK pathway	Inhibits STING expression and activates the AMPK pathway	Improves the survival rates	Animal experiment	[27]
17	milk fat globule epidermal growth factor 8 (MFG-E8)		Inhibits inflammation and promote GPX4 expression	Improves the survival rates	Animal experiment	[28]
18	Mitsugumin-53 (MG-53)		Inhibits inflammation and apoptosis	Improves the survival rates	Animal experiment	[29]
19	Hepatocyte growth factor (HGF)	c-Met	Activates the mTOR pathway	Improves the survival rates	cell experiment	[30]
20	Interleukin-7 (IL-7)	JAK/STAT pathway	Activates the JAK/STAT pathway	Ameliorates lymphatic dysfunction	clinical trials	[31]
21	Interleukin-15 (IL-15)	IL-15R	Inhibits apoptosis	Improves the survival rates	Animal experiment	[32]
22	Interleukin-22 (IL-22)	IL10R/IL-22R	Activates the STAT3 and ATF4-ATG7 pathways	Reduces inflammation	Animal experiment	[33]
23	Interleukin-38 (IL-38)	NLRP3/IL-1βpathway	Inhibits the NLRP3/IL-1β pathway	Improves the survival rates	Animal experiment	[34,35]
24	Fibroblast growth factor 15 (FGF-15)	FGFR4	Exerts anti-inflammatory effects and regulates Treg activity	Improves the survival rates	Animal experiment	[36]
25	Heat shock protein 22 (Hsp22)	AMPK/mTOR pathway	Regulates the AMPK/mTOR pathway	Improves organ damage	Animal experiment	[37]
26	Insulin-like growth factor-1 (IGF-1)	IGF-1R	Inhibits apoptosis	Reduces apoptosis	Animal experiment	[38]
27	Maf 1	NLRP3 promoter region	In activates NLRP3	Reduces apoptosis	cell experiment	[39]
28	Ac2-26	Fpr2	Inhibits inflammation and apoptosis	Improves the survival rates	Animal experiment	[40]
29	Vaspin (Serpin A12)		Inhibits KLK7 expression	Improves the survival rates	Animal experiment	[41]
30	Adiponectin	PI3K/AKT and AMPK/mTOR pathways	Regulates the PI3K/AKT and AMPK/mTOR pathways	Reduces inflammation and liver damage	Animal experiment	[42,43]
31	Irisin	MITOL/GSDMD pathway	Regulates the MITOL/GSDMD pathway	Reduces apoptosis	Animal experiment	[44]
32	Human serum albumin		Improves blood volume	Improves the survival rates	Clinical trials	[45]
33	Selepressin		Maintain MAP	Prognosis did not improve	2b phase III clinical trial	[46,47]
34	B38-CAP		Promotes angiotensin secretion	Improves the survival rates	Animal experiment	[48]
35	Coenzyme Q10		Inhibits inflammation and increases Beclin expression	Reduces liver inflammation and damage	Animal experiment	[49]
36	Erythropoietin		Regulates macrophage	Improves the survival rates	Clinical trials	[50]
37	S100A8	TLR4	Combines with TLR4 to inhibit the inflammatory response	Extends the survival time	Animal experiment	[51]
38	Crotoxin (CTX)		Inhibits inflammation	Improves the survival rates	Animal experiment	[52]
39	Human chorionic gonadotropin		Inhibits the expression of inflammatory factors and chemokines	Ameliorates organ damage	Animal experiment	[53]
40	Bβ15-42		Inhibits inflammation and protects endothelial barrier integrity	Improves liver and lung tissue damage	Animal experiment	[54]
41	Granulocyte–macrophage colony-stimulating factor		Reduces the ratio of Tregs and restores T cell function	Improves inflammatory response	Preclinical studies	[55]
42	Apelin-13	TGF-β1/SMAD and NOX3/ROS/PFKFB3 pathway	Inhibits TGF-β 1/SMAD and NOX3/ROS/PFKFB3 pathways	Reduces lung inflammation and fibrosis	Animal experiment	[56,57]
43	IVIG	Mixed antibody preparation		No significant improvement in 28-day survival	Clinical trials	[58,59]
44	Anti-HMGB1 antibody	HMGB1	Inhibits bacterial translocation and protect intestinal barrier	Improves the survival rates	Animal experiment	[60,61]
45	Atezolizumab	PD-L1	PD-L1 monoclonal antibody	Improve the survival rates	Animal experiment	[62]
46	Vilobelimab	C5a	C5a antibody	No improvement in survival	Phase II clinical trial	[63]
47	Secukinumab	IL-17A	IL-17A antibody	Improves the survival rates	Animal experiment	[64]
48	Adrecizumab (HAM8101)	ADM	ADM antibody	No improvement in survival	Phase II clinical trial	[65]
49	Tocilizumab	IL-6R	IL-6R antibody	Ameliorates lung damage	Animal experiment	[66]
50	miR-25-5p	TXNIP/NLRP3	Negative regulator of TXNIP	Ameliorates brain damage	Animal experiment	[67]
51	miR-214-3p	CTSB	Inhibits CTSB expression	Ameliorates myocardial damage	Animal experiment	[68]
52	miR-340-5p	MyD88	Lowers MD88 level	Ameliorates myocardial damage	Animal experiment	[69]
53	miR-26a-5p	CTGF	Inhibits CTGF expression	Improves the survival rates	Animal experiment	[70]
54	miR-490-3p	RAK1	Inhibits RAK1 expression	Ameliorates lung damage	Animal experiment	[71]
55	miR-164a	Interleukin 1 receptor-associated kinase 1; Tumor necrosis receptor-associated factor 6	Decrease NF-κB activation and splenocyte apoptosis	Ameliorate organ damage	Animal experiment	[72]
56	Heparin	AT	Inhibits clotting and inflammation	Improves the survival rates	Animal experiment	[73]
57	N-acetyl heparin		Inhibits inflammation	Ameliorates lung damage	Animal experiment	[73]
58	Lentinan	NF-κB pathway	Inhibits NF-κB pathway	Ameliorates tissue damage	Animal experiment	[74]
59	Lycium barbarum polysaccharide	PKM2	Promotes ubiquitination of PKM2	Reduces inflammation	Cell experiment	[75]
60	Poria cocos polysaccharide	TLR4	Inhibits NF-κB-p65 phosphorylation	Improves the survival rates	Animal experiment	[76]
61	Tannic acid		Upregulates GABAA receptors expression	Improves anxious behavior	Animal experiment	[77]
62	Eritoran	TLR4	Blocks TLR4 expression	No significant improvement in 28-day survival	Phase III clinical trial	[78]
63	Vasopressin (VP)		Increased mean arterial pressure	No significant improvement in 28-day sur-vival	Clinical trial	[79,80]

ADM: Adrenomedullin; AKT: protein kinase B; AMPK: adenosine 5′-monophosphate (AMP)-activated protein kinase; AT: Antithrombin; ATF4: activating transcription factor 4; ATG7: autophagy related protein 7; C5a: component 5a; c-Met: cellular mesenchymal–epithelial transition factor; CTGF: connective tissue growth factor; CTSB: cathepsin B; FGFR4: fibroblast growth factor receptor 4; Fpr2: Formyl-peptide receptor-2; GABAA: γ-aminobutyric acid sub-type A; GPX4: glutathione peroxidase 4; GSDMD: Adermin D; HMGB1: high-mobility group box protein 1; IL10R: interleukin-10 receptor; IL-15R: interleukin-15 receptor; IL-1β: interleukin-1β; IL-22R: interleukin-22 receptor; IRAK1: interleukin 1 receptor-associated kinase 1; JAK: Janus Kinase; KLK7: kallikrein 7; LPS: lipopolysaccharide; MAP: mean arterial pressure; MAPK: mitogen-activated protein kinase; MITOL: mitochondrial ubiquitin ligase; mTOR: mammalian target of rapamycin; MyD88: myeloid differentiation factor 88; NOX3:NADPH oxidase 3; ROS: reactive oxygen species; NF-κB: nuclear factor kappa-light-chain-enhancer of activated B cells; NLRP3: NOD-like receptor thermal-protein-domain-associated protein 3; NRF2: nuclear factor erythroid 2-related factor 2; HO-1: heme oxygenase-1; PD-L1: programmed cell death-ligand 1; PFKFB3: 6-Phosphofructo-2-kinase/fructose-2,6-bisphosphatase; PI3K: phosphoinositide 3-kinase; PKM2: Pyruvate kinase muscle isoform 2; S6K1: p70 ribosomal protein subunit 6 kinase 1; SMAD: drosophila mothers against decapentaplegic protein; STAT: signal transducer and activator of transcription; STING: stimulator of interferon genes; TATc: thrombin–antithrombin complex; TGF-β1: Transforming growth factor β1; TLR4: Toll-like receptor 4; TXNIP: thioredoxin-interacting protein.

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
