# Peer review of "Research Progress of Macromolecules in the Prevention and Treatment of Sepsis"

_ijms, 2023, doi:10.3390/ijms241613017_

Round 1
Author Response
July 21, 2023
Prof. Dr. Maurizio Battino
Editor-in-Chief
Ms. Maria Otilia Drugan
Assistant Editor
International Journal of Molecular Sciences
Dear Editor,
I wish to re-submit the manuscript titled “Research progress of macromolecules in the prevention and treatment of sepsis.” The manuscript ID is ijms-2513678.
We express our gratitude to you and the reviewers for your valuable suggestions and insightful perspectives. The manuscript has greatly benefited from these astute recommendations.
Attached is the revised version of our manuscript. In the following pages are our point-by-point responses to each of the comments of the reviewers. Revisions in the text are highlighted by the utilization of the color red. We hope that the revisions in the manuscript and our accompanying responses would be sufficient to make our manuscript suitable for publication in International Journal of Molecular Sciences.
Thank you for your consideration. I look forward to hearing from you.
Sincerely,
Jingqian Su, Ph.D.
Associate Professor
Fujian Key Laboratory of Innate Immune Biology
Biomedical Research Center of South China
College of Life Science, Fujian Normal University
Fuzhou 350117, Fujian, China
Tel: +86-18950498937
E-mail: sjq027@fjnu.edu.cn
Responses to the comments of Reviewer #1
- From the manuscript it is not clear whether this is a systematically gathered overview or a summery of molecules chosen by the authors. Why were macromolecules chosen? What is the cut-off for macro and what is the rational of macro versus micro? What is the goal of providing this overview? What is the target audience? Clinicians at the ICU? A lack of effect size and a clearer explanation of the mode of action is missing. Researchers at the lab? Again the effect size is missing as well as comments on side-effects or new developments.
Response:
We wish to extend our sincere gratitude to the reviewers for their invaluable suggestions. In the manuscript, a comprehensive analysis was conducted to thoroughly deliberate upon these matters, including the selection of macromolecules, macro boundaries, and other relevant aspects, while also incorporating justifications. We undertake a systematic review to comprehensively summarize the assortment of both large and small molecule drugs utilized for the management and prophylaxis of sepsis within the time frame spanning from 2012 to 2023. The primary purpose of this article is to provide a comprehensive overview of macromolecules. The laboratory researchers are the main recipients of the findings of our study. The therapeutic mechanism and effect of macromolecules are discussed in detail in the text, with Table 1 providing a comprehensive summary of the research progress in this field. For additional details pertaining to the alteration, please consult lines 42-55 as delineated herein:
Macromolecular drugs (molecular weight ≥ 1000 Da), also known as biological products, are biological macromolecules produced using modern biotechnology for the prevention or treatment of diseases. The FDA categorizes macromolecular drugs into vaccines, blood products, and gene and cell therapy agents based on the nature of drugs. These drugs can include proteins, polypeptides, polysaccharides, antibodies, and nucleic acids. In comparison with small molecule drugs, macromolecular drugs offer advantages such as precise targeting, high efficiency, and long half-life [6,7].
To date, summary articles on the application of macromolecules in sepsis are lacking. Therefore, this review focuses on articles published in PubMed during 2012–2023 that discuss the use of macromolecules in the treatment of sepsis and classify them according to their properties, therapeutic effects, and mechanisms of action. The inflammation detection methods described in these articles involve qPCR and ELISA technologies to measure the mRNA and protein levels of cytokines such as IL-1β, IL-6, and TNF-α among others.
Table 1. Summary of sepsis macromolecules.
Serial number |
Name |
Target/pathway |
Mechanism of action |
Effect |
Research progress |
Reference number |
|
1 |
C1-esterase inhibitor (C1-INH) |
|
Inhibits complement and thrombin activation |
Reduces mortality |
clinical trials |
[8] |
|
2 |
Recombinant antithrombin gamma (RAT-γ) |
|
Inhibits coagulation factor |
Improves patient outcomes |
clinical trials |
[9] |
|
3 |
Recombinant thrombomodulin (rhTM) |
Thrombin |
Improves coagulation function |
Improves blood clotting |
clinical trials |
[10,11] |
|
4 |
Human soluble thrombomodulin (ART-123) |
Thrombin |
Reduces the ratio of D-dimer and TATc |
Improves blood clotting |
clinical trials |
[12] |
|
5 |
Fibroblast growth factor 2 (FGF-2) |
AKT/mTOR/S6K1 pathway |
Suppresses the AKT/mTOR/S6K1 pathway |
Improves blood clotting |
Animal experiment |
[13] |
|
6 |
Annexin A5 |
TLR4 |
Inhibits LPS and HMGB1 binding of the TLR4 |
Improve blood clotting |
Animal experiment |
[14] |
|
7 |
Polymyxin B |
|
Neutralizes endotoxin |
|
clinical trials |
[15,16] |
|
8 |
Secretory leukocyte protease inhibitor (SLPI) |
NF-κB and MAPK pathways |
Inhibits the NF-κB and MAPK pathways |
Inhibit inflammation |
Animal experiment |
[17] |
|
9 |
PS1-2 |
NF-κB and AMPK pathways |
Dissolves the membrane dissolution and inhibits the NF-κB and AMPK pathway |
Inhibits fungal infections |
Animal experiment |
[18] |
|
10 |
KDEON WK-11 |
|
Broad-spectrum antibacterial activity |
Inhibits microbial activity |
Cell experiment |
[19] |
|
11 |
WIKE-14 |
|
Neutralize endotoxin |
Ameliorates LPS-induced acute lung injury |
Animal experiment |
[20] |
|
12 |
Aspidasept (Pep19-2.5) |
NF-κB pathway |
Neutralizes endotoxin and inhibits the NF-κB pathway |
Reduces pyroptosis |
Animal experiment |
[21,22] |
|
13 |
AB103 |
CD28 |
CD28 antagonist |
Ameliorates organ damage |
Phase II clinical trial |
[23,24] |
|
14 |
Alpha-Chymotrypsin (α-ch) |
TLR4/NF-κB pathway |
Inhibits the TLR4/NF-κB pathway |
Improves the survival rates |
Animal experiment |
[25] |
|
15 |
Fibroblast growth factor 19 (FGF-19) |
NRF2/HO-1 pathway |
Activates the NRF2/HO-1 pathway |
Ameliorates organ damage |
Animal experiment |
[26] |
|
16 |
Growth Differentiation Factor 7 GDF-7 |
STING/AMPK pathway |
Inhibits STING expression and activates the AMPK pathway |
Improves the survival rates |
Animal experiment |
[27] |
|
17 |
Milk fat globule growth factor 8 (MFG-E8) |
|
Inhibits inflammation and promote GPX4 expression |
Improves the survival rates |
Animal experiment |
[28] |
|
18 |
Mitsugumin-53 (MG-53) |
|
Inhibits inflammation and apoptosis |
Improves the survival rates |
Animal experiment |
[29] |
|
19 |
Hepatocyte growth factor (HGF) |
c-Met |
Activates the mTOR pathway |
Improves the survival rates |
cell experiment |
[30] |
|
20 |
Interleukin-7 (IL-7) |
JAK/STAT pathway |
Activates the JAK/STAT pathway |
Ameliorates lymphatic dysfunction |
clinical trials |
[31] |
|
21 |
Interleukin-15 (IL-15) |
IL-15R |
Inhibits apoptosis |
Improves the survival rates |
Animal experiment |
[32] |
|
22 |
Interleukin-22 (IL-22) |
IL10R/IL-22R |
Activates the STAT3 and ATF4-ATG7 pathways |
Reduces inflammation |
Animal experiment |
[33] |
|
23 |
Interleukin-38 (IL-38) |
NLRP3/IL-1βpathway |
Inhibits the NLRP3/IL-1β pathway |
Improves the survival rates |
Animal experiment |
[34,35] |
|
24 |
Fibroblast growth factor 15 (FGF-15) |
FGFR4 |
Exerts anti-inflammatory effects and regulates Treg activity |
Improves the survival rates |
Animal experiment |
[36] |
|
25 |
Heat shock protein 22 (Hsp22) |
AMPK/mTOR pathway |
Regulates the AMPK/mTOR pathway |
Improves organ damage |
Animal experiment |
[37] |
|
26 |
Insulin-like growth factor-1 (IGF-1) |
IGF-1R |
Inhibits apoptosis |
Reduces apoptosis |
Animal experiment |
[38] |
|
27 |
Maf 1 |
NLRP3 promoter region |
In activates NLRP3 |
Reduces apoptosis |
cell experiment |
[39] |
|
28 |
Ac2-26 |
Fpr2 |
Inhibits inflammation and apoptosis
|
Improves the survival rates |
Animal experiment |
[40] |
|
29 |
Vaspin (Serpin A12) |
|
Inhibits KLK7 expression |
Improves the survival rates |
Animal experiment |
[41] |
|
30 |
Adiponectin (APN) |
PI3K/AKT and AMPK/mTOR pathways |
Regulates the PI3K/AKT and AMPK/mTOR pathways |
Reduces inflammation and liver damage |
Animal experiment |
[42,43] |
|
31 |
Irisin |
MITOL/GSDMD pathway |
Regulates the MITOL/GSDMD pathway |
Reduces apoptosis |
Animal experiment |
[44] |
|
32 |
Human serum albumin (HSA) |
|
Improves blood volume
|
Improves the survival rates |
clinical trials |
[45] |
|
33 |
Selepressin |
|
Maintain MAP |
Prognosis did not improve |
2b phase III clinical trial |
[46,47] |
|
34 |
B38-CAP |
|
Promotes angiotensin secretion |
Improves the survival rates |
Animal experiment |
[48] |
|
35 |
Coenzyme Q10 |
|
Inhibits inflammation and increases Beclin expression |
Reduces liver inflammation and damage |
Animal experiment |
[49] |
|
36 |
Erythropoietin (EPO) |
|
Regulates macrophage |
Improves the survival rates |
clinical trials |
[50] |
|
37 |
S100A8 |
TLR4 |
Combines with TLR4 to inhibit the inflammatory response |
Extends the survival time |
Animal experiment |
[51] |
|
38 |
Crotoxin (CTX) |
|
Inhibits inflammation
|
Improves the survival rates |
Animal experiment |
[52] |
|
39 |
Human chorionic gonadotropin (HCG) |
|
Inhibits the expression of inflammatory factors and chemokines |
Ameliorates organ damage |
Animal experiment |
[53] |
|
40 |
Bβ15-42 |
|
Inhibits inflammation and protects endothelial barrier integrity |
Improves liver and lung tissue damage |
Animal experiment |
[54] |
|
41 |
Granulocyte–macrophage colony-stimulating factor (GM-CSF) |
|
Reduces the ratio of Tregs and restores T cell function |
Improves inflammatory response |
Preclinical studies |
[55] |
|
42 |
Apelin-13 |
TGF-β1/SMAD and NOX3/ROS/PFKFB3 pathway |
Inhibits TGF-β 1 / SMAD and NOX3/ROS/PFKFB3 pathways
|
Reduces lung inflammation and fibrosis |
Animal experiment |
[56,57] |
|
43 |
IVIG |
Mixed antibody preparation |
|
No significant improvement in 28-day survival |
clinical trials |
[58,59] |
|
44 |
Anti-HMGB1 antibody |
HMGB1 |
Inhibits bacterial translocation and protect intestinal barrier |
Improves the survival rates |
Animal experiment |
[60,61] |
|
45 |
Atezolizumab |
PD-L1 |
PD-L1 monoclonal antibody |
Improve the survival rates |
Animal experiment |
[62] |
|
46 |
Vilobelimab |
C5a |
C5a antibody |
No improvement in survival |
Phase II clinical trial |
[63] |
|
47 |
Secukinumab |
IL-17A |
IL-17A antibody |
Improves the survival rates |
Animal experiment |
[64] |
|
48 |
Adrecizumab (HAM8101) |
ADM |
ADM antibody |
No improvement in survival |
Phase II clinical trial |
[65] |
|
49 |
Tocilizumab |
IL-6R |
IL-6R antibody |
Ameliorates lung damage |
Animal experiment |
[66] |
|
50 |
miR-25-5p |
TXNIP/NLRP3 |
Negative regulator of TXNIP |
Ameliorates brain damage |
Animal experiment |
[67] |
|
51 |
miR-214-3p |
CTSB |
Inhibits CTSB expression |
Ameliorates myocardial damage |
Animal experiment |
[68] |
|
52 |
miR-340-5p |
MyD88 |
Lowers MD88 level |
Ameliorates myocardial damage |
Animal experiment |
[69] |
|
53 |
miR-26a-5p |
CTGF |
Inhibits CTGF expression |
Improves the survival rates |
Animal experiment |
[70] |
|
54 |
miR-490-3p |
RAK1 |
Inhibits RAK1 expression
|
Ameliorates lung damage |
Animal experiment |
[71] |
|
55 |
miR-164a |
Interleukin 1 receptor-associated kinase 1 ï¼›Tumor necrosis receptor-associated factor 6 |
Decrease NF-κB activation and splenocyte apoptosis. |
Ameliorate organ damage |
Animal experiment |
[72] |
|
56 |
Heparin (UFH) |
AT |
Inhibits clotting and inflammation |
Improves the survival rates |
Animal experiment |
[72] |
|
57 |
N-acetyl heparin (NAH) |
|
Inhibits inflammation |
Ameliorates lung damage |
Animal experiment |
[73] |
|
58 |
Lentinan |
NF-κB pathway |
Inhibits NF-κB pathway |
Ameliorates tissue damage |
Animal experiment |
[74] |
|
59 |
Lycium barbarum polysaccharide (LBP) |
PKM2 |
Promotes ubiquitination of PKM2 |
Reduces inflammation |
cell experiment |
[75] |
|
60 |
Poria cocos polysaccharide (PCPs) |
TLR4 |
Inhibits NF-κB-p65 phosphorylation |
Improves the survival rates |
Animal experiment |
[76] |
|
61 |
Tannic acid |
|
Upregulates GABAA expression |
Improves anxious behavior |
Animal experiment |
[77] |
|
62 |
Eritoran |
TLR4 |
Blocks TLR4 expression |
No significant improvement in 28-day survival |
Phase III clinical trial |
[78] |
|
- The authors state the for the treatment of sepsis no specific drugs are available. Antibiotics, some of which are large molecules, and inotropes are not mentioned but very often used in daily practise. Looking at serum albumin, the authors only discuss 1 study. But many more studies have been published on this subject. Why was this article chosen and briefly discussed? How about other molecules? The article only summarizes the molecules as mentioned in the table. A very brief statement about the mechanism of action is given and for some a short statement about efficacy. None of these statements show any effect size, making it hard to judge how clinically relevant the efficacy is.
Response:
We would like to extend our heartfelt appreciation for the invaluable suggestion put forth by the reviewer.
In our investigation of sepsis-related articles from the years 2012-2023, we specifically focused on antibiotics and inotropes. However, our findings were limited to the identification of polymyxin B as the only macromolecule-based antibiotic available; inotropes, such as Dobutamine, Milrinone, and Levosimendan, are classified as small molecular compounds, thus falling beyond the purview of this article.
The articles selected for albumin were chosen based on their relevance to sepsis and updated information. The text has been supplemented with efficacy information, highlighted in red font. Furthermore, the present study incorporates an examination of the therapeutic impact of macromolecules, as outlined below:
(1) Line 68-84
Antithrombin (AT) has anticoagulant and anti-inflammatory effects. Recombi-nant antithrombin gamma (RAT-γ) is a recombinant form of AT lacking fucose. A ret-rospective study of 49 patients with sepsis with antithrombin deficiency and DIC found that the RAT-γ (36 IU/kg) and AT treatment groups showed a significant reduc-tion in JAAM-DIC and SOFA scores in sepsis patients. Additionally, the coagulation disorders and organ failure in these patients were improved [9].
Recombinant thrombomodulin (rhTM) can increase the expression of Glypcan1 and reduce the levels of inflammatory factors in the serum, improve cardiac microcir-culation disorder and vascular injury caused by sepsis, and enhance the 48-h survival rate of lipopolysaccharide (LPS)-induced sepsis mouse models. A retrospective study on patients with DIC revealed that rhTM treatment significantly reduced the DIC score, CRP levels, and SOFA score, improving coagulation function, inflammatory re-sponse, and organ damage [10,11].
Human soluble thrombomodulin (ART-123) is a recombinant soluble form of hu-man thrombomodulin. The multinational SCARLET phase III study included 800 pa-tients and reported that ART-123 effectively improved abnormalities in patients with sepsis-associated coagulation disease, but did not significantly improve the 28-day survival rate [12].
(2) Line 106-115
Polymyxin B (PMB) exerts both antibacterial and neutralizing functions. A pro-spective cohort study involving 60 patients with abdominal infectious sepsis found that PMB effectively reduced the level of endotoxin activity in septic shock patients and significantly improved their SOFA scores, but not survival [15]. Additionally, a meta-analysis of 12,234 critically ill adult patients showed that PMB has a higher probability of causing nephrotoxicity in patients [16].
The PS1-2 peptide has shown enhanced antifungal activity through membrane dissolution and inhibited TLR2 activity and TNF-a expression levels in RAW264. It ef-fectively combat sexually transmitted infections and inflammatory responses without triggering drug resistance [18].
(3) Line 124-130
Aspidasept (Pep19-2.5) is an anti-lipopolysaccharide polypeptide with a high binding affinity to LPS, neutralizes endotoxins, and exerts anti-inflammatory effects. In vitro cell and animal model toxicity tests demonstrate its safety since the therapeutic dose (30–50 μg/kg) is significantly lower than the toxic dose (2 mg/kg). Pep19-2.5 can reduce LPS-induced pyroptosis by preventing LPS from binding to TLR4, inhibiting the activation of NF-κB pathway, reducing the inflammatory response caused by LPS, and inhibiting the activation of caspase-1[21,22].
(4) Line 140-162
Alpha-Chymotrypsin (α-ch) is a serine protease. By inhibiting the TLR4/NF-κB pathway, α-ch, a serine protease, reduces the levels of inflammatory cytokines IL-1β, IL-6, and TNF-α and the expression of MPO and iNOS in a cecum ligation and punc-ture (CLP)-induced sepsis mouse model. After α-ch treatment, the survival rate of CLP mice increased by 50% [25].
Fibroblast growth factor 19 (FGF-19) can reduce the expression of LA and GLA, activate the NRF2/HO-1 pathway, and inhibit the expression of INOS, ROS, caspase-3, and cytochrome c, thereby alleviating lipid metabolism disorders, liver injury, and kidney injury in LPS-induced sepsis mice [26].
Growth Differentiation Factor 7 (GDF7) can reduce inflammation and oxidative stress by downregulating STING-activated AMPK signaling, improving LPS-induced lung injury in septic mice; the 72-h survival rate of septic mice treated with GDF7 ex-ceeds 50% [27].
As a biomarker of sepsis, milk fat globule epidermal growth factor 8 (MFG-E8) significantly reduces the levels of inflammatory factors and the concentration of MDA and ferrous ions in the liver, increased the ratio of GSH and GPX4 in the antioxidant system, and inhibits oxidative stress and ferroptosis in the liver of CLP mouse model. MFG-E8 has a protective effect on liver injury and increases the survival rate of CLP mice by >40% [28].
Mitsugumin-53 (MG53) is a protein of the TRIM family that exhibits an-ti-inflammatory and anti-oxidative properties, promotes plasma membrane repair, and improves myocardial dysfunction caused by sepsis by upregulating PPARα ex-pression. Additionally, treatment with MG53 has been shown to increase the 72-h sur-vival rate (>60%) of CLP rats [29].
(5) Line 198-202
IL-38 is a cytokine of the IL-1 family that inhibits the NLRP3/IL-1β signaling pathway and Treg cell immune function, reduces the expression of inflammatory mediators and pro-apoptotic proteins, attenuates early inflammatory response and apoptosis, improves the depletion of effector T cells during sepsis, and increases the survival rate of septic mice (>60%) [34,35].
(6) Line 212-216
Ac2-26 is an N-terminal active peptide of AnnexinA1 that targets Fpr2 receptors and inhibits PI3K and AKT phosphorylation. Ac2-26 downregulates the level of NF-κB, reduces the expression of inflammatory cytokines and pro-apoptotic proteins in septic mice, inhibits inflammation and apoptosis, improves renal injury, and increases the 7-day survival rate of CLP mice (>40%) [40].
(7) Line 250-263
HSA regulates the blood osmotic pressure [85] and improves edema, hypoalbu-minemia, and hypovolemia caused by sepsis. A retrospective study involving 5,009 pa-tients with sepsis demonstrated that treatment with 5% HSA significantly reduced sepsis-related 28-day mortality (8.7%) [45].
Selepressin is a selective vasopressin V1a agonist. A phase IIa randomized, place-bo-controlled trial with 53 patients with septic shock showed that Selepressin (2.5 ng/kg/min) could replace norepinephrine (18.24 μg/kg/min) to maintain mean arterial pressure balance with good tolerance [46]. However, a phase III/IIb clinical study with 868 patients with septic shock revealed that Selepressin did not significantly reduce the usage of ventilators and pressor drugs in patients with septic shock within 30 days [47].
B38-CAP, an ACE2-like enzyme derived from Bacillus-like bacteria, can upregu-late the level of angiotensin 1–7, reduce the levels of AngII and inflammatory cyto-kines and chemokines, improve lung injury caused by sepsis, and increase the survival rate of CLP mice by 20% [48].
(8) Line 330-334
IVIG is a mixed IgG antibody. A single-center retrospective study of 239 patients with sepsis showed that IVIG significantly improved the prognosis of patients with low IgG levels and reduced mortality. A retrospective multicenter study of 850 pa-tients with severe COVID-19 showed that IVIG did not significantly improve the 28-day survival rate of patients, and might have induced adverse thrombosis and al-lergic reactions [58,59].
(9) Line 345-360
Vilobelimab is a monoclonal antibody against the allergic toxin C5a that amelio-rates infectious organ dysfunction by neutralizing C5a. A randomized, double-blind, multicenter phase IIa study of 72 patients with sepsis or septic shock showed that vi-lobelimab neutralizes C5a in a dose-dependent manner with good tolerance and safety. However, the study did not show significant improvement in sepsis prognosis and mortality [63].
Secukinumab is a human monoclonal antibody that neutralizes IL-17A to inhibit the IKB/NF-κB pathway, reduce the expression of pro-inflammatory factors, such as IL-6 and TNF-α, and improve the inflammatory response and lung injury induced by sepsis. Simultaneously, the survival rate of CLP rats injected with a high-dose secuki-numab (20 mg/kg) was 60% after 196 h [64].
Adrecizumab (HAM8101) is a non-neutralizing human adrenomedullin mono-clonal antibody that reduces the expression of VEGF, increases the level of Angiopoi-etin-1, improves the effect of sepsis on vascular permeability, and reduces inflamma-tion and mortality in septic mice [65]. . A phase 2, double-blind, randomized, con-trolled trial involving 301 patients with septic shock found that adrecizumab was well tolerated but did not significantly improve the survival rate of patients [88].
(10) Line 398-403
Furthermore, the inhibition of the NF-κB pathway through the targeting of IRAK1/TRAF6 by miR-146a has been observed to effectively reduce the inflammatory response in mice subjected to CLP, as well as mitigate splenocyte apoptosis. Notably, the administration of miR-146a resulted in a significant improvement in sep-sis-induced organ damage, leading to a noteworthy 40% increase in the survival rate of CLP mice (n=15) [72].
(11) Line 437-441
LBP is the primary bioactive component of the traditional Chinese medicine Lycium barbarum and exerts anti-inflammation and antioxidation effects. LBP promotes PKM2 ubiquitination by increasing the expression of Nedd4L, Nedd4, and Gnb2 in Raw264.7 cells, reduces the level of PKM2, and inhibits LPS-induced glycolysis process and cell differentiation of M1 macrophages. LBP also reduces the expression of IL-1β, TNF-α, and HMGB1 and alleviates the inflammatory response induced by LPS [75].
3.The conclusion section start of with new information, making it not a conclusion based on the data presented. The authors suggest to optimize research. What is meant by this statement?
Response:
We express our sincere gratitude for the reviewer's invaluable suggestion. In accordance with the reviewer's perspective, a new section has been integrated into the revised manuscript (lines 485-502) as outlined below:
Compared with small molecules, biomacromolecules have advantages such as precise targeting, long half-life, low toxicity, and fewer side effects. Their large spatial structure enables more stable target binding. However, biomacromolecules also have several shortcomings. In comparison with small molecules, the preparation process of biomacromolecules is immature. Most biomacromolecules, such as protein and nucleic acid drugs, require low-temperature storage to prevent degradation and have higher requirements for storage and transportation equipment. Biomacromolecules, owing to their larger molecular weight, are difficult to be absorbed and are prone to immuno-genicity. Additionally, the biological activity of biomacromolecules such as proteins, polypeptides, and nucleic acids relies on their spatial structure, making oral admin-istration challenging. Therefore, optimizing the research and development process of molecular preparation to reduce production costs and improve product purity is es-sential. Currently, there are relatively few immunogenicity challenges, such as the humanization of antibodies. Overcoming the challenges of absorption can be ad-dressed by identifying therapeutic fragments of proteins and peptides, reducing mo-lecular weight, and enhancing absorption efficiency through nanoparticle conjugation or cooperation with cell-penetrating peptides. Coating methods and other approaches can also facilitate the oral administration of macromolecules [6,7,94].
- The authors propose that macromolecules have good prospects. Where is this proposition based upon?
Response:
We would like to extend our heartfelt appreciation for the invaluable suggestion offered by the reviewer. The promising potential of macromolecules has been incorporated into the revised manuscript (line 503-509), as follow:
Macromolecular drugs demonstrate superior therapeutic efficacy, tolerability, and safety. The therapeutic dose of certain macromolecules is lower than that of small molecules, and macromolecules have a longer half-life, which enhances the therapeu-tic effect. Resistance to treatment with antimicrobial peptide macromolecules has also been reported. Furthermore, the therapeutic dose of macromolecular drugs is signifi-cantly lower than their toxic dose, indicating a higher level of safety. Therefore, we be-lieve that macromolecular drugs hold great promise in the treatment of sepsis.

Reviewer 2 Report
Thank you so much for reviewing a very informative and well-written manuscript. Why don't you separately record the cell experiments, animal experiments, and clinical trials in the main text? I would like the outcomes of the study to be included in Table 1.

I found a few minor errors with regard to word spacing.
Author Response
July 21, 2023
Prof. Dr. Maurizio Battino
Editor-in-Chief
Ms. Maria Otilia Drugan
Assistant Editor
International Journal of Molecular Sciences
Dear Editor,
I wish to re-submit the manuscript titled “Research progress of macromolecules in the prevention and treatment of sepsis.” The manuscript ID is ijms-2513678.
We express our gratitude to you and the reviewers for your valuable suggestions and insightful perspectives. The manuscript has greatly benefited from these astute recommendations.
Attached is the revised version of our manuscript. In the following pages are our point-by-point responses to each of the comments of the reviewers. Revisions in the text are highlighted by the utilization of the color red. We hope that the revisions in the manuscript and our accompanying responses would be sufficient to make our manuscript suitable for publication in International Journal of Molecular Sciences.
Thank you for your consideration. I look forward to hearing from you.
Sincerely,
Jingqian Su, Ph.D.
Associate Professor
Fujian Key Laboratory of Innate Immune Biology
Biomedical Research Center of South China
College of Life Science, Fujian Normal University
Fuzhou 350117, Fujian, China
Tel: +86-18950498937
E-mail: sjq027@fjnu.edu.cn
Responses to the comments of Reviewer #2
- Thank you so much for reviewing a very informative and well-written manuscript. Why don't you separately record the cell experiments, animal experiments, and clinical trials in the main text? I would like the outcomes of the study to be included in Table 1.
Response:
Many thanks for the reviewer suggestion. We are grateful for the suggestion. The separate descriptions of clinical trials, animal experiments, and cell experiments are presented in Table 1, as follow:
Table 1. Summary of sepsis macromolecules.
Serial number |
Name |
Target/pathway |
Mechanism of action |
Effect |
Research progress |
Reference number |
|
1 |
C1-esterase inhibitor (C1-INH) |
|
Inhibits complement and thrombin activation |
Reduces mortality |
clinical trials |
[8] |
|
2 |
Recombinant antithrombin gamma (RAT-γ) |
|
Inhibits coagulation factor |
Improves patient outcomes |
clinical trials |
[9] |
|
3 |
Recombinant thrombomodulin (rhTM) |
Thrombin |
Improves coagulation function |
Improves blood clotting |
clinical trials |
[10,11] |
|
4 |
Human soluble thrombomodulin (ART-123) |
Thrombin |
Reduces the ratio of D-dimer and TATc |
Improves blood clotting |
clinical trials |
[12] |
|
5 |
Fibroblast growth factor 2 (FGF-2) |
AKT/mTOR/S6K1 pathway |
Suppresses the AKT/mTOR/S6K1 pathway |
Improves blood clotting |
Animal experiment |
[13] |
|
6 |
Annexin A5 |
TLR4 |
Inhibits LPS and HMGB1 binding of the TLR4 |
Improve blood clotting |
Animal experiment |
[14] |
|
7 |
Polymyxin B |
|
Neutralizes endotoxin |
|
clinical trials |
[15,16] |
|
8 |
Secretory leukocyte protease inhibitor (SLPI) |
NF-κB and MAPK pathways |
Inhibits the NF-κB and MAPK pathways |
Inhibit inflammation |
Animal experiment |
[17] |
|
9 |
PS1-2 |
NF-κB and AMPK pathways |
Dissolves the membrane dissolution and inhibits the NF-κB and AMPK pathway |
Inhibits fungal infections |
Animal experiment |
[18] |
|
10 |
KDEON WK-11 |
|
Broad-spectrum antibacterial activity |
Inhibits microbial activity |
Cell experiment |
[19] |
|
11 |
WIKE-14 |
|
Neutralize endotoxin |
Ameliorates LPS-induced acute lung injury |
Animal experiment |
[20] |
|
12 |
Aspidasept (Pep19-2.5) |
NF-κB pathway |
Neutralizes endotoxin and inhibits the NF-κB pathway |
Reduces pyroptosis |
Animal experiment |
[21,22] |
|
13 |
AB103 |
CD28 |
CD28 antagonist |
Ameliorates organ damage |
Phase II clinical trial |
[23,24] |
|
14 |
Alpha-Chymotrypsin (α-ch) |
TLR4/NF-κB pathway |
Inhibits the TLR4/NF-κB pathway |
Improves the survival rates |
Animal experiment |
[25] |
|
15 |
Fibroblast growth factor 19 (FGF-19) |
NRF2/HO-1 pathway |
Activates the NRF2/HO-1 pathway |
Ameliorates organ damage |
Animal experiment |
[26] |
|
16 |
Growth Differentiation Factor 7 GDF-7 |
STING/AMPK pathway |
Inhibits STING expression and activates the AMPK pathway |
Improves the survival rates |
Animal experiment |
[27] |
|
17 |
Milk fat globule growth factor 8 (MFG-E8) |
|
Inhibits inflammation and promote GPX4 expression |
Improves the survival rates |
Animal experiment |
[28] |
|
18 |
Mitsugumin-53 (MG-53) |
|
Inhibits inflammation and apoptosis |
Improves the survival rates |
Animal experiment |
[29] |
|
19 |
Hepatocyte growth factor (HGF) |
c-Met |
Activates the mTOR pathway |
Improves the survival rates |
cell experiment |
[30] |
|
20 |
Interleukin-7 (IL-7) |
JAK/STAT pathway |
Activates the JAK/STAT pathway |
Ameliorates lymphatic dysfunction |
clinical trials |
[31] |
|
21 |
Interleukin-15 (IL-15) |
IL-15R |
Inhibits apoptosis |
Improves the survival rates |
Animal experiment |
[32] |
|
22 |
Interleukin-22 (IL-22) |
IL10R/IL-22R |
Activates the STAT3 and ATF4-ATG7 pathways |
Reduces inflammation |
Animal experiment |
[33] |
|
23 |
Interleukin-38 (IL-38) |
NLRP3/IL-1βpathway |
Inhibits the NLRP3/IL-1β pathway |
Improves the survival rates |
Animal experiment |
[34,35] |
|
24 |
Fibroblast growth factor 15 (FGF-15) |
FGFR4 |
Exerts anti-inflammatory effects and regulates Treg activity |
Improves the survival rates |
Animal experiment |
[36] |
|
25 |
Heat shock protein 22 (Hsp22) |
AMPK/mTOR pathway |
Regulates the AMPK/mTOR pathway |
Improves organ damage |
Animal experiment |
[37] |
|
26 |
Insulin-like growth factor-1 (IGF-1) |
IGF-1R |
Inhibits apoptosis |
Reduces apoptosis |
Animal experiment |
[38] |
|
27 |
Maf 1 |
NLRP3 promoter region |
In activates NLRP3 |
Reduces apoptosis |
cell experiment |
[39] |
|
28 |
Ac2-26 |
Fpr2 |
Inhibits inflammation and apoptosis
|
Improves the survival rates |
Animal experiment |
[40] |
|
29 |
Vaspin (Serpin A12) |
|
Inhibits KLK7 expression |
Improves the survival rates |
Animal experiment |
[41] |
|
30 |
Adiponectin (APN) |
PI3K/AKT and AMPK/mTOR pathways |
Regulates the PI3K/AKT and AMPK/mTOR pathways |
Reduces inflammation and liver damage |
Animal experiment |
[42,43] |
|
31 |
Irisin |
MITOL/GSDMD pathway |
Regulates the MITOL/GSDMD pathway |
Reduces apoptosis |
Animal experiment |
[44] |
|
32 |
Human serum albumin (HSA) |
|
Improves blood volume
|
Improves the survival rates |
clinical trials |
[45] |
|
33 |
Selepressin |
|
Maintain MAP |
Prognosis did not improve |
2b phase III clinical trial |
[46,47] |
|
34 |
B38-CAP |
|
Promotes angiotensin secretion |
Improves the survival rates |
Animal experiment |
[48] |
|
35 |
Coenzyme Q10 |
|
Inhibits inflammation and increases Beclin expression |
Reduces liver inflammation and damage |
Animal experiment |
[49] |
|
36 |
Erythropoietin (EPO) |
|
Regulates macrophage |
Improves the survival rates |
clinical trials |
[50] |
|
37 |
S100A8 |
TLR4 |
Combines with TLR4 to inhibit the inflammatory response |
Extends the survival time |
Animal experiment |
[51] |
|
38 |
Crotoxin (CTX) |
|
Inhibits inflammation
|
Improves the survival rates |
Animal experiment |
[52] |
|
39 |
Human chorionic gonadotropin (HCG) |
|
Inhibits the expression of inflammatory factors and chemokines |
Ameliorates organ damage |
Animal experiment |
[53] |
|
40 |
Bβ15-42 |
|
Inhibits inflammation and protects endothelial barrier integrity |
Improves liver and lung tissue damage |
Animal experiment |
[54] |
|
41 |
Granulocyte–macrophage colony-stimulating factor (GM-CSF) |
|
Reduces the ratio of Tregs and restores T cell function |
Improves inflammatory response |
Preclinical studies |
[55] |
|
42 |
Apelin-13 |
TGF-β1/SMAD and NOX3/ROS/PFKFB3 pathway |
Inhibits TGF-β 1 / SMAD and NOX3/ROS/PFKFB3 pathways
|
Reduces lung inflammation and fibrosis |
Animal experiment |
[56,57] |
|
43 |
IVIG |
Mixed antibody preparation |
|
No significant improvement in 28-day survival |
clinical trials |
[58,59] |
|
44 |
Anti-HMGB1 antibody |
HMGB1 |
Inhibits bacterial translocation and protect intestinal barrier |
Improves the survival rates |
Animal experiment |
[60,61] |
|
45 |
Atezolizumab |
PD-L1 |
PD-L1 monoclonal antibody |
Improve the survival rates |
Animal experiment |
[62] |
|
46 |
Vilobelimab |
C5a |
C5a antibody |
No improvement in survival |
Phase II clinical trial |
[63] |
|
47 |
Secukinumab |
IL-17A |
IL-17A antibody |
Improves the survival rates |
Animal experiment |
[64] |
|
48 |
Adrecizumab (HAM8101) |
ADM |
ADM antibody |
No improvement in survival |
Phase II clinical trial |
[65] |
|
49 |
Tocilizumab |
IL-6R |
IL-6R antibody |
Ameliorates lung damage |
Animal experiment |
[66] |
|
50 |
miR-25-5p |
TXNIP/NLRP3 |
Negative regulator of TXNIP |
Ameliorates brain damage |
Animal experiment |
[67] |
|
51 |
miR-214-3p |
CTSB |
Inhibits CTSB expression |
Ameliorates myocardial damage |
Animal experiment |
[68] |
|
52 |
miR-340-5p |
MyD88 |
Lowers MD88 level |
Ameliorates myocardial damage |
Animal experiment |
[69] |
|
53 |
miR-26a-5p |
CTGF |
Inhibits CTGF expression |
Improves the survival rates |
Animal experiment |
[70] |
|
54 |
miR-490-3p |
RAK1 |
Inhibits RAK1 expression
|
Ameliorates lung damage |
Animal experiment |
[71] |
|
55 |
miR-164a |
Interleukin 1 receptor-associated kinase 1 ï¼›Tumor necrosis receptor-associated factor 6 |
Decrease NF-κB activation and splenocyte apoptosis. |
Ameliorate organ damage |
Animal experiment |
[72] |
|
56 |
Heparin (UFH) |
AT |
Inhibits clotting and inflammation |
Improves the survival rates |
Animal experiment |
[72] |
|
57 |
N-acetyl heparin (NAH) |
|
Inhibits inflammation |
Ameliorates lung damage |
Animal experiment |
[73] |
|
58 |
Lentinan |
NF-κB pathway |
Inhibits NF-κB pathway |
Ameliorates tissue damage |
Animal experiment |
[74] |
|
59 |
Lycium barbarum polysaccharide (LBP) |
PKM2 |
Promotes ubiquitination of PKM2 |
Reduces inflammation |
cell experiment |
[75] |
|
60 |
Poria cocos polysaccharide (PCPs) |
TLR4 |
Inhibits NF-κB-p65 phosphorylation |
Improves the survival rates |
Animal experiment |
[76] |
|
61 |
Tannic acid |
|
Upregulates GABAA expression |
Improves anxious behavior |
Animal experiment |
[77] |
|
62 |
Eritoran |
TLR4 |
Blocks TLR4 expression |
No significant improvement in 28-day survival |
Phase III clinical trial |
[78] |
|

Reviewer 3 Report
Authors tried to review the macromolecules in the prevention and treatment of sepsis. There is no specific drug effective in sepsis over the world. In that sense, it could be helpful to review the macromolecules investigated in sepsis. However, the followings should be discussed.
Major issue
There should be more detailed and updated information. For example, in line 29, they commented that afelimomab and fucoflavin have been used for the treatment of sepsis, but it is not true. It is not used in sepsis management in clinical medicine.
Likewise, the outcomes of clinical trials about eritoran should be included. Also, many drugs investigated in animal study have been failure in clinical trials, which needs comments.
Minor issues
In line 51, they said macromolecular drugs currently used in the treatment of sepsis, but it is not true. The sepsis guideline does not endorse any drug.
In table 1, there are many acronyms, and they should be spelled out.
In table 1, selepressin maintains MAP, not MPA. It seems to be typo.
Also there are some typos and grammatical errors.
In line 153, typo should be corrected.
In line 187, selenium should be changed to selepressin.
GM-CSF is in category of anti-inflammation. It is immune-enhancing drug, rather than anti-inflammatory drug.
In figure 7. miR-49-3p might be typo.
miR-146a is also well known to regulate IRAK-1.
Author Response
July 21, 2023
Prof. Dr. Maurizio Battino
Editor-in-Chief
Ms. Maria Otilia Drugan
Assistant Editor
International Journal of Molecular Sciences
Dear Editor,
I wish to re-submit the manuscript titled “Research progress of macromolecules in the prevention and treatment of sepsis.” The manuscript ID is ijms-2513678.
We express our gratitude to you and the reviewers for your valuable suggestions and insightful perspectives. The manuscript has greatly benefited from these astute recommendations.
Attached is the revised version of our manuscript. In the following pages are our point-by-point responses to each of the comments of the reviewers. Revisions in the text are highlighted by the utilization of the color red. We hope that the revisions in the manuscript and our accompanying responses would be sufficient to make our manuscript suitable for publication in International Journal of Molecular Sciences.
Thank you for your consideration. I look forward to hearing from you.
Sincerely,
Jingqian Su, Ph.D.
Associate Professor
Fujian Key Laboratory of Innate Immune Biology
Biomedical Research Center of South China
College of Life Science, Fujian Normal University
Fuzhou 350117, Fujian, China
Tel: +86-18950498937
E-mail: sjq027@fjnu.edu.cn
Responses to the comments of Reviewer #3
- There should be more detailed and updated information. For example, in line 29, they
commented that afelimomab and fucoflavin have been used for the treatment of sepsis, but it is not true. It is not used in sepsis management in clinical medicine.
Response:
We would like to extend our heartfelt appreciation for the invaluable suggestion put forth by the reviewer. According to the review comments, we have extensively discussed the aforementioned suggestion, resulting in the implementation of specific modifications (lines 37-41). The contents are as follows:
Currently, both large and small molecules have been used for the treatment and preven-tion of sepsis such as human serum albumin (HSA) and antibiotics. Additionally, labora-tory research suggests that afelimomab, fucoxanthin, and the fermented drink kombucha significantly improve sepsis-induced symptoms, particularly by reducing the inflamma-tory response [3–5].
- Likewise, the outcomes of clinical trials about eritoran should be included. Also, many drugs investigated in animal study have been failure in clinical trials, which needs comments.
Response:
We express our gratitude to the reviewers for their invaluable comments, which have greatly assisted us. A comprehensive analysis is provided on the pharmaceuticals that have exhibited clinical inefficacy. The contents are as follows:
(1) Line 81-84
Human soluble thrombomodulin (ART-123) is a recombinant soluble form of hu-man thrombomodulin. The multinational SCARLET phase III study included 800 pa-tients and reported that ART-123 effectively improved abnormalities in patients with sepsis-associated coagulation disease, but did not significantly improve the 28-day survival rate [12].
(2)Line 106-111
Polymyxin B (PMB) exerts both antibacterial and neutralizing functions. A pro-spective cohort study involving 60 patients with abdominal infectious sepsis found that PMB effectively reduced the level of endotoxin activity in septic shock patients and significantly improved their SOFA scores, but not survival [15]. Additionally, a meta-analysis of 12,234 critically ill adult patients showed that PMB has a higher probability of causing nephrotoxicity in patients [16].
(3)Line 254-259
Selepressin is a selective vasopressin V1a agonist. A phase IIa randomized, place-bo-controlled trial with 53 patients with septic shock showed that Selepressin (2.5 ng/kg/min) could replace norepinephrine (18.24 μg/kg/min) to maintain mean arterial pressure balance with good tolerance [46]. However, a phase III/IIb clinical study with 868 patients with septic shock revealed that Selepressin did not significantly reduce the usage of ventilators and pressor drugs in patients with septic shock within 30 days [47].
(4)Line 308-312
AB103 is a CD28 antagonist that inhibits inflammation and improves the effi-ciency of bacterial clearance in a bacterial sepsis model. A multicenter, randomized clinical trial involving 40 patients with NSTI showed that AB103 could improve organ damage and safety. However, it did not significantly affect the duration of ventilator use or the level of cytokines [23,24].
(5)Line 330-334
IVIG is a mixed IgG antibody. A single-center retrospective study of 239 patients with sepsis showed that IVIG significantly improved the prognosis of patients with low IgG levels and reduced mortality. A retrospective multicenter study of 850 pa-tients with severe COVID-19 showed that IVIG did not significantly improve the 28-day survival rate of patients, and might have induced adverse thrombosis and al-lergic reactions [58,59].
(6)Line 345-349
Vilobelimab is a monoclonal antibody against the allergic toxin C5a that amelio-rates infectious organ dysfunction by neutralizing C5a. A randomized, double-blind, multicenter phase IIa study of 72 patients with sepsis or septic shock showed that vi-lobelimab neutralizes C5a in a dose-dependent manner with good tolerance and safety. However, the study did not show significant improvement in sepsis prognosis and mortality [63].
(7)Line 355-360
Adrecizumab (HAM8101) is a non-neutralizing human adrenomedullin mono-clonal antibody that reduces the expression of VEGF, increases the level of Angiopoi-etin-1, improves the effect of sepsis on vascular permeability, and reduces inflamma-tion and mortality in septic mice [65]. . A phase 2, double-blind, randomized, con-trolled trial involving 301 patients with septic shock found that adrecizumab was well tolerated but did not significantly improve the survival rate of patients [88].
(8)Line 423-430
UFH, a glycosaminoglycan, and NAH, and non-anticoagulant, can reduce lung leukocyte infiltration, capillary barrier injury, and pulmonary edema in septic mice, inhibit caspase-11-induced cell pyrolysis induced by LPS and HMGB1, reduce the in-crease of pro-inflammatory cytokines IL-1 β and TNF-α, and inhibit HPA activity to protect endothelial glycocalyx. The anti-inflammatory effect of UFH is superior to that of NAH; however, UFH increases the risk of bleeding. A retrospective study of 3,377 SIC patients with sepsis found that UFH (6250-13750 IU/d) reduced the risk of death in patients with an SIC score of 4; however, UFH increased the risk of bleeding in clinical trial [73,91,92].
(9) Line 456-460
Eritoran is a derivative of lipopolysaccharide (LPS) lipid A and a TLR4 blocker. Eritoran can inhibit the expression of inflammatory mediators, inhibit the cytokine storm in sepsis, increase Th1 reaction time, and reduce death in mice with bacterial sepsis. However, in a randomized, double-blind, phase 3 clinical trial of 1,961 subjects, eritoran could not reduce the 28-day mortality in patients with severe sepsis [78,93].
- In line 51, they said macromolecular drugs currently used in the treatment of sepsis,
but it is not true. The sepsis guideline does not endorse any drug.
Response:
We express our gratitude for your valuable comment and extend our sincere apologies for any confusion that may have arisen. The paragraph (line 62-63) has been revised, as follow:
Macromolecular drugs are presently employed in clinical trials and animal experiments for the treatment of sepsis, such as rhTM), ART-123, FGF-2, and An-nexin A5, have anti-coagulant functions (Figure 1).
- In table 1, there are many acronyms, and they should be spelled out. In table 1, selepressin maintains MAP, not MPA.
Response:
We express our sincere gratitude for the valuable suggestion provided by the reviewer. The complete designation of the abbreviation has been included in Table 1. Selepressin maintains MAP, corrected. The contents are as follows:
Table 1. Summary of sepsis macromolecules.
Serial number |
Name |
Target/pathway |
Mechanism of action |
Effect |
Research progress |
Reference number |
|
1 |
C1-esterase inhibitor (C1-INH) |
|
Inhibits complement and thrombin activation |
Reduces mortality |
clinical trials |
[8] |
|
2 |
Recombinant antithrombin gamma (RAT-γ) |
|
Inhibits coagulation factor |
Improves patient outcomes |
clinical trials |
[9] |
|
3 |
Recombinant thrombomodulin (rhTM) |
Thrombin |
Improves coagulation function |
Improves blood clotting |
clinical trials |
[10,11] |
|
4 |
Human soluble thrombomodulin (ART-123) |
Thrombin |
Reduces the ratio of D-dimer and TATc |
Improves blood clotting |
clinical trials |
[12] |
|
5 |
Fibroblast growth factor 2 (FGF-2) |
AKT/mTOR/S6K1 pathway |
Suppresses the AKT/mTOR/S6K1 pathway |
Improves blood clotting |
Animal experiment |
[13] |
|
6 |
Annexin A5 |
TLR4 |
Inhibits LPS and HMGB1 binding of the TLR4 |
Improve blood clotting |
Animal experiment |
[14] |
|
7 |
Polymyxin B |
|
Neutralizes endotoxin |
|
clinical trials |
[15,16] |
|
8 |
Secretory leukocyte protease inhibitor (SLPI) |
NF-κB and MAPK pathways |
Inhibits the NF-κB and MAPK pathways |
Inhibit inflammation |
Animal experiment |
[17] |
|
9 |
PS1-2 |
NF-κB and AMPK pathways |
Dissolves the membrane dissolution and inhibits the NF-κB and AMPK pathway |
Inhibits fungal infections |
Animal experiment |
[18] |
|
10 |
KDEON WK-11 |
|
Broad-spectrum antibacterial activity |
Inhibits microbial activity |
Cell experiment |
[19] |
|
11 |
WIKE-14 |
|
Neutralize endotoxin |
Ameliorates LPS-induced acute lung injury |
Animal experiment |
[20] |
|
12 |
Aspidasept (Pep19-2.5) |
NF-κB pathway |
Neutralizes endotoxin and inhibits the NF-κB pathway |
Reduces pyroptosis |
Animal experiment |
[21,22] |
|
13 |
AB103 |
CD28 |
CD28 antagonist |
Ameliorates organ damage |
Phase II clinical trial |
[23,24] |
|
14 |
Alpha-Chymotrypsin (α-ch) |
TLR4/NF-κB pathway |
Inhibits the TLR4/NF-κB pathway |
Improves the survival rates |
Animal experiment |
[25] |
|
15 |
Fibroblast growth factor 19 (FGF-19) |
NRF2/HO-1 pathway |
Activates the NRF2/HO-1 pathway |
Ameliorates organ damage |
Animal experiment |
[26] |
|
16 |
Growth Differentiation Factor 7 GDF-7 |
STING/AMPK pathway |
Inhibits STING expression and activates the AMPK pathway |
Improves the survival rates |
Animal experiment |
[27] |
|
17 |
Milk fat globule growth factor 8 (MFG-E8) |
|
Inhibits inflammation and promote GPX4 expression |
Improves the survival rates |
Animal experiment |
[28] |
|
18 |
Mitsugumin-53 (MG-53) |
|
Inhibits inflammation and apoptosis |
Improves the survival rates |
Animal experiment |
[29] |
|
19 |
Hepatocyte growth factor (HGF) |
c-Met |
Activates the mTOR pathway |
Improves the survival rates |
cell experiment |
[30] |
|
20 |
Interleukin-7 (IL-7) |
JAK/STAT pathway |
Activates the JAK/STAT pathway |
Ameliorates lymphatic dysfunction |
clinical trials |
[31] |
|
21 |
Interleukin-15 (IL-15) |
IL-15R |
Inhibits apoptosis |
Improves the survival rates |
Animal experiment |
[32] |
|
22 |
Interleukin-22 (IL-22) |
IL10R/IL-22R |
Activates the STAT3 and ATF4-ATG7 pathways |
Reduces inflammation |
Animal experiment |
[33] |
|
23 |
Interleukin-38 (IL-38) |
NLRP3/IL-1βpathway |
Inhibits the NLRP3/IL-1β pathway |
Improves the survival rates |
Animal experiment |
[34,35] |
|
24 |
Fibroblast growth factor 15 (FGF-15) |
FGFR4 |
Exerts anti-inflammatory effects and regulates Treg activity |
Improves the survival rates |
Animal experiment |
[36] |
|
25 |
Heat shock protein 22 (Hsp22) |
AMPK/mTOR pathway |
Regulates the AMPK/mTOR pathway |
Improves organ damage |
Animal experiment |
[37] |
|
26 |
Insulin-like growth factor-1 (IGF-1) |
IGF-1R |
Inhibits apoptosis |
Reduces apoptosis |
Animal experiment |
[38] |
|
27 |
Maf 1 |
NLRP3 promoter region |
In activates NLRP3 |
Reduces apoptosis |
cell experiment |
[39] |
|
28 |
Ac2-26 |
Fpr2 |
Inhibits inflammation and apoptosis
|
Improves the survival rates |
Animal experiment |
[40] |
|
29 |
Vaspin (Serpin A12) |
|
Inhibits KLK7 expression |
Improves the survival rates |
Animal experiment |
[41] |
|
30 |
Adiponectin (APN) |
PI3K/AKT and AMPK/mTOR pathways |
Regulates the PI3K/AKT and AMPK/mTOR pathways |
Reduces inflammation and liver damage |
Animal experiment |
[42,43] |
|
31 |
Irisin |
MITOL/GSDMD pathway |
Regulates the MITOL/GSDMD pathway |
Reduces apoptosis |
Animal experiment |
[44] |
|
32 |
Human serum albumin (HSA) |
|
Improves blood volume
|
Improves the survival rates |
clinical trials |
[45] |
|
33 |
Selepressin |
|
Maintain MAP |
Prognosis did not improve |
2b phase III clinical trial |
[46,47] |
|
34 |
B38-CAP |
|
Promotes angiotensin secretion |
Improves the survival rates |
Animal experiment |
[48] |
|
35 |
Coenzyme Q10 |
|
Inhibits inflammation and increases Beclin expression |
Reduces liver inflammation and damage |
Animal experiment |
[49] |
|
36 |
Erythropoietin (EPO) |
|
Regulates macrophage |
Improves the survival rates |
clinical trials |
[50] |
|
37 |
S100A8 |
TLR4 |
Combines with TLR4 to inhibit the inflammatory response |
Extends the survival time |
Animal experiment |
[51] |
|
38 |
Crotoxin (CTX) |
|
Inhibits inflammation
|
Improves the survival rates |
Animal experiment |
[52] |
|
39 |
Human chorionic gonadotropin (HCG) |
|
Inhibits the expression of inflammatory factors and chemokines |
Ameliorates organ damage |
Animal experiment |
[53] |
|
40 |
Bβ15-42 |
|
Inhibits inflammation and protects endothelial barrier integrity |
Improves liver and lung tissue damage |
Animal experiment |
[54] |
|
41 |
Granulocyte–macrophage colony-stimulating factor (GM-CSF) |
|
Reduces the ratio of Tregs and restores T cell function |
Improves inflammatory response |
Preclinical studies |
[55] |
|
42 |
Apelin-13 |
TGF-β1/SMAD and NOX3/ROS/PFKFB3 pathway |
Inhibits TGF-β 1 / SMAD and NOX3/ROS/PFKFB3 pathways
|
Reduces lung inflammation and fibrosis |
Animal experiment |
[56,57] |
|
43 |
IVIG |
Mixed antibody preparation |
|
No significant improvement in 28-day survival |
clinical trials |
[58,59] |
|
44 |
Anti-HMGB1 antibody |
HMGB1 |
Inhibits bacterial translocation and protect intestinal barrier |
Improves the survival rates |
Animal experiment |
[60,61] |
|
45 |
Atezolizumab |
PD-L1 |
PD-L1 monoclonal antibody |
Improve the survival rates |
Animal experiment |
[62] |
|
46 |
Vilobelimab |
C5a |
C5a antibody |
No improvement in survival |
Phase II clinical trial |
[63] |
|
47 |
Secukinumab |
IL-17A |
IL-17A antibody |
Improves the survival rates |
Animal experiment |
[64] |
|
48 |
Adrecizumab (HAM8101) |
ADM |
ADM antibody |
No improvement in survival |
Phase II clinical trial |
[65] |
|
49 |
Tocilizumab |
IL-6R |
IL-6R antibody |
Ameliorates lung damage |
Animal experiment |
[66] |
|
50 |
miR-25-5p |
TXNIP/NLRP3 |
Negative regulator of TXNIP |
Ameliorates brain damage |
Animal experiment |
[67] |
|
51 |
miR-214-3p |
CTSB |
Inhibits CTSB expression |
Ameliorates myocardial damage |
Animal experiment |
[68] |
|
52 |
miR-340-5p |
MyD88 |
Lowers MD88 level |
Ameliorates myocardial damage |
Animal experiment |
[69] |
|
53 |
miR-26a-5p |
CTGF |
Inhibits CTGF expression |
Improves the survival rates |
Animal experiment |
[70] |
|
54 |
miR-490-3p |
RAK1 |
Inhibits RAK1 expression
|
Ameliorates lung damage |
Animal experiment |
[71] |
|
55 |
miR-164a |
Interleukin 1 receptor-associated kinase 1 ï¼›Tumor necrosis receptor-associated factor 6 |
Decrease NF-κB activation and splenocyte apoptosis. |
Ameliorate organ damage |
Animal experiment |
[72] |
|
56 |
Heparin (UFH) |
AT |
Inhibits clotting and inflammation |
Improves the survival rates |
Animal experiment |
[72] |
|
57 |
N-acetyl heparin (NAH) |
|
Inhibits inflammation |
Ameliorates lung damage |
Animal experiment |
[73] |
|
58 |
Lentinan |
NF-κB pathway |
Inhibits NF-κB pathway |
Ameliorates tissue damage |
Animal experiment |
[74] |
|
59 |
Lycium barbarum polysaccharide (LBP) |
PKM2 |
Promotes ubiquitination of PKM2 |
Reduces inflammation |
cell experiment |
[75] |
|
60 |
Poria cocos polysaccharide (PCPs) |
TLR4 |
Inhibits NF-κB-p65 phosphorylation |
Improves the survival rates |
Animal experiment |
[76] |
|
61 |
Tannic acid |
|
Upregulates GABAA expression |
Improves anxious behavior |
Animal experiment |
[77] |
|
62 |
Eritoran |
TLR4 |
Blocks TLR4 expression |
No significant improvement in 28-day survival |
Phase III clinical trial |
[78] |
|
5.It seems to be typo. Also there are some typos and grammatical errors. In line 153, typo should be corrected . In line 187, selenium should be changed to selepressin.
Response:
We hereby extend our gratitude for your perceptive commentary and sincerely apologize for any potential misinterpretations that may have arisen. The typos of lines 153 and 187 are corrected (line 203-205 and 248), as follows:
(1) Line 203-205
Fibroblast growth factor 15 (FGF-15) is an analog of FGF-19 FGF-15 that signifi-cantly improves liver inflammation and apoptosis in septic mice by targeting FGFR4, reducing the proportion of Tregs, and increasing the survival rate of septic mice by 40% [36].
(2) Line 248
HAS, Selepressin, and B38-CAP improve blood volume and hypotension during sepsis treatment.
- GM-CSF is in category of anti-inflammation. It is immune-enhancing drug, rather
than anti-inflammatory drug.
Response:
We express our sincere gratitude for the reviewer's invaluable suggestion. The statement regarding the function and mechanism of GM-CSF has been rectified (line 292-294 and Table 1).
(1) line 292-294
Granulocyte-macrophage colony-stimulating factor (GM-CSF) is a glycoprotein. In patients with liver cirrhosis and sepsis, GM-CSF therapy regulates the ratio of MDSCs and Tregs and improves immune function of CD4+ T cells [55].
(2) Table 1
Table 1. Summary of sepsis macromolecules.
Serial number |
Name |
Target/pathway |
Mechanism of action |
Effect |
Research progress |
Reference number |
|
1 |
C1-esterase inhibitor (C1-INH) |
|
Inhibits complement and thrombin activation |
Reduces mortality |
clinical trials |
[8] |
|
2 |
Recombinant antithrombin gamma (RAT-γ) |
|
Inhibits coagulation factor |
Improves patient outcomes |
clinical trials |
[9] |
|
3 |
Recombinant thrombomodulin (rhTM) |
Thrombin |
Improves coagulation function |
Improves blood clotting |
clinical trials |
[10,11] |
|
4 |
Human soluble thrombomodulin (ART-123) |
Thrombin |
Reduces the ratio of D-dimer and TATc |
Improves blood clotting |
clinical trials |
[12] |
|
5 |
Fibroblast growth factor 2 (FGF-2) |
AKT/mTOR/S6K1 pathway |
Suppresses the AKT/mTOR/S6K1 pathway |
Improves blood clotting |
Animal experiment |
[13] |
|
6 |
Annexin A5 |
TLR4 |
Inhibits LPS and HMGB1 binding of the TLR4 |
Improve blood clotting |
Animal experiment |
[14] |
|
7 |
Polymyxin B |
|
Neutralizes endotoxin |
|
clinical trials |
[15,16] |
|
8 |
Secretory leukocyte protease inhibitor (SLPI) |
NF-κB and MAPK pathways |
Inhibits the NF-κB and MAPK pathways |
Inhibit inflammation |
Animal experiment |
[17] |
|
9 |
PS1-2 |
NF-κB and AMPK pathways |
Dissolves the membrane dissolution and inhibits the NF-κB and AMPK pathway |
Inhibits fungal infections |
Animal experiment |
[18] |
|
10 |
KDEON WK-11 |
|
Broad-spectrum antibacterial activity |
Inhibits microbial activity |
Cell experiment |
[19] |
|
11 |
WIKE-14 |
|
Neutralize endotoxin |
Ameliorates LPS-induced acute lung injury |
Animal experiment |
[20] |
|
12 |
Aspidasept (Pep19-2.5) |
NF-κB pathway |
Neutralizes endotoxin and inhibits the NF-κB pathway |
Reduces pyroptosis |
Animal experiment |
[21,22] |
|
13 |
AB103 |
CD28 |
CD28 antagonist |
Ameliorates organ damage |
Phase II clinical trial |
[23,24] |
|
14 |
Alpha-Chymotrypsin (α-ch) |
TLR4/NF-κB pathway |
Inhibits the TLR4/NF-κB pathway |
Improves the survival rates |
Animal experiment |
[25] |
|
15 |
Fibroblast growth factor 19 (FGF-19) |
NRF2/HO-1 pathway |
Activates the NRF2/HO-1 pathway |
Ameliorates organ damage |
Animal experiment |
[26] |
|
16 |
Growth Differentiation Factor 7 GDF-7 |
STING/AMPK pathway |
Inhibits STING expression and activates the AMPK pathway |
Improves the survival rates |
Animal experiment |
[27] |
|
17 |
Milk fat globule growth factor 8 (MFG-E8) |
|
Inhibits inflammation and promote GPX4 expression |
Improves the survival rates |
Animal experiment |
[28] |
|
18 |
Mitsugumin-53 (MG-53) |
|
Inhibits inflammation and apoptosis |
Improves the survival rates |
Animal experiment |
[29] |
|
19 |
Hepatocyte growth factor (HGF) |
c-Met |
Activates the mTOR pathway |
Improves the survival rates |
cell experiment |
[30] |
|
20 |
Interleukin-7 (IL-7) |
JAK/STAT pathway |
Activates the JAK/STAT pathway |
Ameliorates lymphatic dysfunction |
clinical trials |
[31] |
|
21 |
Interleukin-15 (IL-15) |
IL-15R |
Inhibits apoptosis |
Improves the survival rates |
Animal experiment |
[32] |
|
22 |
Interleukin-22 (IL-22) |
IL10R/IL-22R |
Activates the STAT3 and ATF4-ATG7 pathways |
Reduces inflammation |
Animal experiment |
[33] |
|
23 |
Interleukin-38 (IL-38) |
NLRP3/IL-1βpathway |
Inhibits the NLRP3/IL-1β pathway |
Improves the survival rates |
Animal experiment |
[34,35] |
|
24 |
Fibroblast growth factor 15 (FGF-15) |
FGFR4 |
Exerts anti-inflammatory effects and regulates Treg activity |
Improves the survival rates |
Animal experiment |
[36] |
|
25 |
Heat shock protein 22 (Hsp22) |
AMPK/mTOR pathway |
Regulates the AMPK/mTOR pathway |
Improves organ damage |
Animal experiment |
[37] |
|
26 |
Insulin-like growth factor-1 (IGF-1) |
IGF-1R |
Inhibits apoptosis |
Reduces apoptosis |
Animal experiment |
[38] |
|
27 |
Maf 1 |
NLRP3 promoter region |
In activates NLRP3 |
Reduces apoptosis |
cell experiment |
[39] |
|
28 |
Ac2-26 |
Fpr2 |
Inhibits inflammation and apoptosis
|
Improves the survival rates |
Animal experiment |
[40] |
|
29 |
Vaspin (Serpin A12) |
|
Inhibits KLK7 expression |
Improves the survival rates |
Animal experiment |
[41] |
|
30 |
Adiponectin (APN) |
PI3K/AKT and AMPK/mTOR pathways |
Regulates the PI3K/AKT and AMPK/mTOR pathways |
Reduces inflammation and liver damage |
Animal experiment |
[42,43] |
|
31 |
Irisin |
MITOL/GSDMD pathway |
Regulates the MITOL/GSDMD pathway |
Reduces apoptosis |
Animal experiment |
[44] |
|
32 |
Human serum albumin (HSA) |
|
Improves blood volume
|
Improves the survival rates |
clinical trials |
[45] |
|
33 |
Selepressin |
|
Maintain MAP |
Prognosis did not improve |
2b phase III clinical trial |
[46,47] |
|
34 |
B38-CAP |
|
Promotes angiotensin secretion |
Improves the survival rates |
Animal experiment |
[48] |
|
35 |
Coenzyme Q10 |
|
Inhibits inflammation and increases Beclin expression |
Reduces liver inflammation and damage |
Animal experiment |
[49] |
|
36 |
Erythropoietin (EPO) |
|
Regulates macrophage |
Improves the survival rates |
clinical trials |
[50] |
|
37 |
S100A8 |
TLR4 |
Combines with TLR4 to inhibit the inflammatory response |
Extends the survival time |
Animal experiment |
[51] |
|
38 |
Crotoxin (CTX) |
|
Inhibits inflammation
|
Improves the survival rates |
Animal experiment |
[52] |
|
39 |
Human chorionic gonadotropin (HCG) |
|
Inhibits the expression of inflammatory factors and chemokines |
Ameliorates organ damage |
Animal experiment |
[53] |
|
40 |
Bβ15-42 |
|
Inhibits inflammation and protects endothelial barrier integrity |
Improves liver and lung tissue damage |
Animal experiment |
[54] |
|
41 |
Granulocyte–macrophage colony-stimulating factor (GM-CSF) |
|
Reduces the ratio of Tregs and restores T cell function |
Improves inflammatory response |
Preclinical studies |
[55] |
|
42 |
Apelin-13 |
TGF-β1/SMAD and NOX3/ROS/PFKFB3 pathway |
Inhibits TGF-β 1 / SMAD and NOX3/ROS/PFKFB3 pathways
|
Reduces lung inflammation and fibrosis |
Animal experiment |
[56,57] |
|
43 |
IVIG |
Mixed antibody preparation |
|
No significant improvement in 28-day survival |
clinical trials |
[58,59] |
|
44 |
Anti-HMGB1 antibody |
HMGB1 |
Inhibits bacterial translocation and protect intestinal barrier |
Improves the survival rates |
Animal experiment |
[60,61] |
|
45 |
Atezolizumab |
PD-L1 |
PD-L1 monoclonal antibody |
Improve the survival rates |
Animal experiment |
[62] |
|
46 |
Vilobelimab |
C5a |
C5a antibody |
No improvement in survival |
Phase II clinical trial |
[63] |
|
47 |
Secukinumab |
IL-17A |
IL-17A antibody |
Improves the survival rates |
Animal experiment |
[64] |
|
48 |
Adrecizumab (HAM8101) |
ADM |
ADM antibody |
No improvement in survival |
Phase II clinical trial |
[65] |
|
49 |
Tocilizumab |
IL-6R |
IL-6R antibody |
Ameliorates lung damage |
Animal experiment |
[66] |
|
50 |
miR-25-5p |
TXNIP/NLRP3 |
Negative regulator of TXNIP |
Ameliorates brain damage |
Animal experiment |
[67] |
|
51 |
miR-214-3p |
CTSB |
Inhibits CTSB expression |
Ameliorates myocardial damage |
Animal experiment |
[68] |
|
52 |
miR-340-5p |
MyD88 |
Lowers MD88 level |
Ameliorates myocardial damage |
Animal experiment |
[69] |
|
53 |
miR-26a-5p |
CTGF |
Inhibits CTGF expression |
Improves the survival rates |
Animal experiment |
[70] |
|
54 |
miR-490-3p |
RAK1 |
Inhibits RAK1 expression
|
Ameliorates lung damage |
Animal experiment |
[71] |
|
55 |
miR-164a |
Interleukin 1 receptor-associated kinase 1 ï¼›Tumor necrosis receptor-associated factor 6 |
Decrease NF-κB activation and splenocyte apoptosis. |
Ameliorate organ damage |
Animal experiment |
[72] |
|
56 |
Heparin (UFH) |
AT |
Inhibits clotting and inflammation |
Improves the survival rates |
Animal experiment |
[72] |
|
57 |
N-acetyl heparin (NAH) |
|
Inhibits inflammation |
Ameliorates lung damage |
Animal experiment |
[73] |
|
58 |
Lentinan |
NF-κB pathway |
Inhibits NF-κB pathway |
Ameliorates tissue damage |
Animal experiment |
[74] |
|
59 |
Lycium barbarum polysaccharide (LBP) |
PKM2 |
Promotes ubiquitination of PKM2 |
Reduces inflammation |
cell experiment |
[75] |
|
60 |
Poria cocos polysaccharide (PCPs) |
TLR4 |
Inhibits NF-κB-p65 phosphorylation |
Improves the survival rates |
Animal experiment |
[76] |
|
61 |
Tannic acid |
|
Upregulates GABAA expression |
Improves anxious behavior |
Animal experiment |
[77] |
|
62 |
Eritoran |
TLR4 |
Blocks TLR4 expression |
No significant improvement in 28-day survival |
Phase III clinical trial |
[78] |
|
- In figure 7. miR-49-3p might be typo. miR-146a is also well known to regulate IRAK-1.
Response:
We would like to extend our heartfelt appreciation for the invaluable suggestion provided by the reviewer.
The miR-49-3p has been corrected to miR-490-3p in manuscript (line 395 and 398) and Figure 7.
The regulatory functions of miR-146a have been added to the manuscript (line 398-403), as follow:
(1) Line 395 and 398
By targeting IRAK1, miR-490-3p inhibits the IRAK1/TRAF6 pathway, reduces the level of NF-κB phosphorylation; decreases the expression of inflammatory factors IL-1β, IL-6, and TNF α in ALI rat model; and improves LPS-induced pulmonary inflammation and apoptosis in rats [71].
(2) Line 398-403
Furthermore, the inhibition of the NF-κB pathway through the targeting of IRAK1/TRAF6 by miR-146a has been observed to effectively reduce the inflammatory response in mice subjected to CLP, as well as mitigate splenocyte apoptosis. Notably, the administration of miR-146a resulted in a significant improvement in sepsis-induced organ damage, leading to a noteworthy 40% increase in the survival rate of CLP mice (n=15) [72].
(3) Figure 7
Figure 7. Therapeutic mechanism of miR-25-5p, miR-214-3p, miR-142-5p, miR-340-5p, miR-26a-5p, and miR-490-3p. miR: microRNA; TXNIP: thioredoxin interacting protein; NLRP3: nucleotide-binding oligomerization domain; CTSB: cathepsin B; Bax: BCL2-Associated
X; ROS: reactive oxygen species; MDA: malondialdehyde; SOD: superoxide dismutase; bcl-2: B-cell lymphoma-2; NF-κB: nuclear factor kappa-light-chain-enhancer of activated B cells; MAPK: mitogen-activated protein kinase; GSH: glutathione; MyD88: myeloid differentiation primary response gene 88; CTGF: connective tissue growth factor; IL-6: interleukin-6; TNF-α: tumor necrosis factor; IRAK1: interleukin 1 receptor associated kinase 1; TRAF6: TNF receptor associated factor 6.

Reviewer 4 Report
Daer auyhors, I have read with great interest your work. However, I have some concerns.
1. please provide the definition of Sepsis (see sepsis-3 and the guidelines provided by the Surviving sepsis campaign)
2. please specify well established and reccomended treatment for sepsis such as source control, antibiotics, and so on.
3. in table 1 all abbreviations should be specified, as also in text
4. All the figures need to be fully explained, with abbreviations but also mechanism of action of different macromolecules (inhibition? stimulation?) It is not clear.
5. The major limit in my opinion is that your paper provides a long list of macromolecules tested in sepsis, but it does not specify the real context (sepsis, septic shock?), the trial type (retospective study, randomized?), the sample size (humans, animals?)the effect (mortality? inflammation? how inflammation was detected?) the real clinical effect, the mechanis, and so on.
Moreover, it does not critically discuss and consider all the limits of different molecules, with different actions, with different populations, with different evidences.
You analyze vasopressors, anticoagulants, antibodies, and so on but they have different mechanism with different indications during sepsis and septic shock. No mention about this important concern.
Also conclusion sare misleading in my opinion, since there is no mention of all these important limits.
-
Author Response
July 21, 2023
Prof. Dr. Maurizio Battino
Editor-in-Chief
Ms. Maria Otilia Drugan
Assistant Editor
International Journal of Molecular Sciences
Dear Editor,
I wish to re-submit the manuscript titled “Research progress of macromolecules in the prevention and treatment of sepsis.” The manuscript ID is ijms-2513678.
We express our gratitude to you and the reviewers for your valuable suggestions and insightful perspectives. The manuscript has greatly benefited from these astute recommendations.
Attached is the revised version of our manuscript. In the following pages are our point-by-point responses to each of the comments of the reviewers. Revisions in the text are highlighted by the utilization of the color red. We hope that the revisions in the manuscript and our accompanying responses would be sufficient to make our manuscript suitable for publication in International Journal of Molecular Sciences.
Thank you for your consideration. I look forward to hearing from you.
Sincerely,
Jingqian Su, Ph.D.
Associate Professor
Fujian Key Laboratory of Innate Immune Biology
Biomedical Research Center of South China
College of Life Science, Fujian Normal University
Fuzhou 350117, Fujian, China
Tel: +86-18950498937
E-mail: sjq027@fjnu.edu.cn
Responses to the comments of Reviewer #4
- please provide the definition of Sepsis (see sepsis-3 and the guidelines provided by the Surviving sepsis campaign)
Response:
Many thanks for the reviewer suggestion. We are grateful for the suggestion. The definition of sepsis has been incorporated (line 21-22), as follows:
Sepsis is a life-threatening organ dysfunction caused by a host response disorder resulting from infection [1]
- please specify well established and reccomended treatment for sepsis such as source control, antibiotics, and so on.
Response:
We express our sincere gratitude for the valuable suggestion provided by the reviewer. We have incorporated established and recommended sepsis treatments into the manuscript (lines 28-37), as follow:
Currently, there is no specific drug for the treatment of sepsis. In clinical practice, patients with infection or suspected infection with a Sequential Organ Failure Assessment (SOFA) score of ≥2 are diagnosed with sepsis, and antibacterial drug treatment should be initiated within 3 h. Obvious infection sites may require surgical management. Anti-inflammatory and immunomodulatory drugs are used to improve the inflammatory response in patients. For patients with disseminated intravascular coagulation (DIC), timely anticoagulant therapy is necessary. Mechanical ventilation and other supportive treatments are required for patients with acute respiratory distress syndrome. Fluid resuscitation should be performed for shock patients. Vasoactive drugs are administered to maintain a mean arterial pressure ≥65 mmHg [1–3]. Currently, both large and small molecules have been used for the treatment and prevention of sepsis such as human serum albumin (HSA) and antibiotics. Additionally, laboratory research suggests that afelimomab, fucoxanthin, and the fermented drink kombucha significantly improve sepsis-induced symptoms, particularly by reducing the inflammatory response [3–5].
- In table 1 all abbreviations should be specified, as also in text.
Response:
We express our sincere gratitude for your valuable comment. All abbreviations in Table 1 and text have been appropriately annotated with their corresponding full names.
Table 1. Summary of sepsis macromolecules.
Serial number |
Name |
Target/pathway |
Mechanism of action |
Effect |
Research progress |
Reference number |
|
1 |
C1-esterase inhibitor (C1-INH) |
|
Inhibits complement and thrombin activation |
Reduces mortality |
clinical trials |
[8] |
|
2 |
Recombinant antithrombin gamma (RAT-γ) |
|
Inhibits coagulation factor |
Improves patient outcomes |
clinical trials |
[9] |
|
3 |
Recombinant thrombomodulin (rhTM) |
Thrombin |
Improves coagulation function |
Improves blood clotting |
clinical trials |
[10,11] |
|
4 |
Human soluble thrombomodulin (ART-123) |
Thrombin |
Reduces the ratio of D-dimer and TATc |
Improves blood clotting |
clinical trials |
[12] |
|
5 |
Fibroblast growth factor 2 (FGF-2) |
AKT/mTOR/S6K1 pathway |
Suppresses the AKT/mTOR/S6K1 pathway |
Improves blood clotting |
Animal experiment |
[13] |
|
6 |
Annexin A5 |
TLR4 |
Inhibits LPS and HMGB1 binding of the TLR4 |
Improve blood clotting |
Animal experiment |
[14] |
|
7 |
Polymyxin B |
|
Neutralizes endotoxin |
|
clinical trials |
[15,16] |
|
8 |
Secretory leukocyte protease inhibitor (SLPI) |
NF-κB and MAPK pathways |
Inhibits the NF-κB and MAPK pathways |
Inhibit inflammation |
Animal experiment |
[17] |
|
9 |
PS1-2 |
NF-κB and AMPK pathways |
Dissolves the membrane dissolution and inhibits the NF-κB and AMPK pathway |
Inhibits fungal infections |
Animal experiment |
[18] |
|
10 |
KDEON WK-11 |
|
Broad-spectrum antibacterial activity |
Inhibits microbial activity |
Cell experiment |
[19] |
|
11 |
WIKE-14 |
|
Neutralize endotoxin |
Ameliorates LPS-induced acute lung injury |
Animal experiment |
[20] |
|
12 |
Aspidasept (Pep19-2.5) |
NF-κB pathway |
Neutralizes endotoxin and inhibits the NF-κB pathway |
Reduces pyroptosis |
Animal experiment |
[21,22] |
|
13 |
AB103 |
CD28 |
CD28 antagonist |
Ameliorates organ damage |
Phase II clinical trial |
[23,24] |
|
14 |
Alpha-Chymotrypsin (α-ch) |
TLR4/NF-κB pathway |
Inhibits the TLR4/NF-κB pathway |
Improves the survival rates |
Animal experiment |
[25] |
|
15 |
Fibroblast growth factor 19 (FGF-19) |
NRF2/HO-1 pathway |
Activates the NRF2/HO-1 pathway |
Ameliorates organ damage |
Animal experiment |
[26] |
|
16 |
Growth Differentiation Factor 7 GDF-7 |
STING/AMPK pathway |
Inhibits STING expression and activates the AMPK pathway |
Improves the survival rates |
Animal experiment |
[27] |
|
17 |
Milk fat globule growth factor 8 (MFG-E8) |
|
Inhibits inflammation and promote GPX4 expression |
Improves the survival rates |
Animal experiment |
[28] |
|
18 |
Mitsugumin-53 (MG-53) |
|
Inhibits inflammation and apoptosis |
Improves the survival rates |
Animal experiment |
[29] |
|
19 |
Hepatocyte growth factor (HGF) |
c-Met |
Activates the mTOR pathway |
Improves the survival rates |
cell experiment |
[30] |
|
20 |
Interleukin-7 (IL-7) |
JAK/STAT pathway |
Activates the JAK/STAT pathway |
Ameliorates lymphatic dysfunction |
clinical trials |
[31] |
|
21 |
Interleukin-15 (IL-15) |
IL-15R |
Inhibits apoptosis |
Improves the survival rates |
Animal experiment |
[32] |
|
22 |
Interleukin-22 (IL-22) |
IL10R/IL-22R |
Activates the STAT3 and ATF4-ATG7 pathways |
Reduces inflammation |
Animal experiment |
[33] |
|
23 |
Interleukin-38 (IL-38) |
NLRP3/IL-1βpathway |
Inhibits the NLRP3/IL-1β pathway |
Improves the survival rates |
Animal experiment |
[34,35] |
|
24 |
Fibroblast growth factor 15 (FGF-15) |
FGFR4 |
Exerts anti-inflammatory effects and regulates Treg activity |
Improves the survival rates |
Animal experiment |
[36] |
|
25 |
Heat shock protein 22 (Hsp22) |
AMPK/mTOR pathway |
Regulates the AMPK/mTOR pathway |
Improves organ damage |
Animal experiment |
[37] |
|
26 |
Insulin-like growth factor-1 (IGF-1) |
IGF-1R |
Inhibits apoptosis |
Reduces apoptosis |
Animal experiment |
[38] |
|
27 |
Maf 1 |
NLRP3 promoter region |
In activates NLRP3 |
Reduces apoptosis |
cell experiment |
[39] |
|
28 |
Ac2-26 |
Fpr2 |
Inhibits inflammation and apoptosis
|
Improves the survival rates |
Animal experiment |
[40] |
|
29 |
Vaspin (Serpin A12) |
|
Inhibits KLK7 expression |
Improves the survival rates |
Animal experiment |
[41] |
|
30 |
Adiponectin (APN) |
PI3K/AKT and AMPK/mTOR pathways |
Regulates the PI3K/AKT and AMPK/mTOR pathways |
Reduces inflammation and liver damage |
Animal experiment |
[42,43] |
|
31 |
Irisin |
MITOL/GSDMD pathway |
Regulates the MITOL/GSDMD pathway |
Reduces apoptosis |
Animal experiment |
[44] |
|
32 |
Human serum albumin (HSA) |
|
Improves blood volume
|
Improves the survival rates |
clinical trials |
[45] |
|
33 |
Selepressin |
|
Maintain MAP |
Prognosis did not improve |
2b phase III clinical trial |
[46,47] |
|
34 |
B38-CAP |
|
Promotes angiotensin secretion |
Improves the survival rates |
Animal experiment |
[48] |
|
35 |
Coenzyme Q10 |
|
Inhibits inflammation and increases Beclin expression |
Reduces liver inflammation and damage |
Animal experiment |
[49] |
|
36 |
Erythropoietin (EPO) |
|
Regulates macrophage |
Improves the survival rates |
clinical trials |
[50] |
|
37 |
S100A8 |
TLR4 |
Combines with TLR4 to inhibit the inflammatory response |
Extends the survival time |
Animal experiment |
[51] |
|
38 |
Crotoxin (CTX) |
|
Inhibits inflammation
|
Improves the survival rates |
Animal experiment |
[52] |
|
39 |
Human chorionic gonadotropin (HCG) |
|
Inhibits the expression of inflammatory factors and chemokines |
Ameliorates organ damage |
Animal experiment |
[53] |
|
40 |
Bβ15-42 |
|
Inhibits inflammation and protects endothelial barrier integrity |
Improves liver and lung tissue damage |
Animal experiment |
[54] |
|
41 |
Granulocyte–macrophage colony-stimulating factor (GM-CSF) |
|
Reduces the ratio of Tregs and restores T cell function |
Improves inflammatory response |
Preclinical studies |
[55] |
|
42 |
Apelin-13 |
TGF-β1/SMAD and NOX3/ROS/PFKFB3 pathway |
Inhibits TGF-β 1 / SMAD and NOX3/ROS/PFKFB3 pathways
|
Reduces lung inflammation and fibrosis |
Animal experiment |
[56,57] |
|
43 |
IVIG |
Mixed antibody preparation |
|
No significant improvement in 28-day survival |
clinical trials |
[58,59] |
|
44 |
Anti-HMGB1 antibody |
HMGB1 |
Inhibits bacterial translocation and protect intestinal barrier |
Improves the survival rates |
Animal experiment |
[60,61] |
|
45 |
Atezolizumab |
PD-L1 |
PD-L1 monoclonal antibody |
Improve the survival rates |
Animal experiment |
[62] |
|
46 |
Vilobelimab |
C5a |
C5a antibody |
No improvement in survival |
Phase II clinical trial |
[63] |
|
47 |
Secukinumab |
IL-17A |
IL-17A antibody |
Improves the survival rates |
Animal experiment |
[64] |
|
48 |
Adrecizumab (HAM8101) |
ADM |
ADM antibody |
No improvement in survival |
Phase II clinical trial |
[65] |
|
49 |
Tocilizumab |
IL-6R |
IL-6R antibody |
Ameliorates lung damage |
Animal experiment |
[66] |
|
50 |
miR-25-5p |
TXNIP/NLRP3 |
Negative regulator of TXNIP |
Ameliorates brain damage |
Animal experiment |
[67] |
|
51 |
miR-214-3p |
CTSB |
Inhibits CTSB expression |
Ameliorates myocardial damage |
Animal experiment |
[68] |
|
52 |
miR-340-5p |
MyD88 |
Lowers MD88 level |
Ameliorates myocardial damage |
Animal experiment |
[69] |
|
53 |
miR-26a-5p |
CTGF |
Inhibits CTGF expression |
Improves the survival rates |
Animal experiment |
[70] |
|
54 |
miR-490-3p |
RAK1 |
Inhibits RAK1 expression
|
Ameliorates lung damage |
Animal experiment |
[71] |
|
55 |
miR-164a |
Interleukin 1 receptor-associated kinase 1 ï¼›Tumor necrosis receptor-associated factor 6 |
Decrease NF-κB activation and splenocyte apoptosis. |
Ameliorate organ damage |
Animal experiment |
[72] |
|
56 |
Heparin (UFH) |
AT |
Inhibits clotting and inflammation |
Improves the survival rates |
Animal experiment |
[72] |
|
57 |
N-acetyl heparin (NAH) |
|
Inhibits inflammation |
Ameliorates lung damage |
Animal experiment |
[73] |
|
58 |
Lentinan |
NF-κB pathway |
Inhibits NF-κB pathway |
Ameliorates tissue damage |
Animal experiment |
[74] |
|
59 |
Lycium barbarum polysaccharide (LBP) |
PKM2 |
Promotes ubiquitination of PKM2 |
Reduces inflammation |
cell experiment |
[75] |
|
60 |
Poria cocos polysaccharide (PCPs) |
TLR4 |
Inhibits NF-κB-p65 phosphorylation |
Improves the survival rates |
Animal experiment |
[76] |
|
61 |
Tannic acid |
|
Upregulates GABAA expression |
Improves anxious behavior |
Animal experiment |
[77] |
|
62 |
Eritoran |
TLR4 |
Blocks TLR4 expression |
No significant improvement in 28-day survival |
Phase III clinical trial |
[78] |
|
- All the figures need to be fully explained, with abbreviations but also mechanism of action of different macromolecules (inhibition? stimulation?) It is not clear.
Response:
We express our sincere gratitude for your valuable comment. we have diligently incorporated the complete nomenclature of the abbreviation within each corresponding legend. Furthermore, to elucidate the precise functioning of macromolecules depicted in the diagram, the activation and inhibition processes have been duly annotated within each diagram. The details are as follows:
Figure 1. Anticoagulation mechanism of rhTM, ART-123, Annexin, and FGF-2. rhTM: Recombinant thrombomodulin; ART-123: Human soluble thrombomodulin; APC: activated protein C; Va: ac-tivated factor five; Vâ…¢a: , activated factor seven; HMGB1: High mobility group box-1 protein; LPS: Lipopolysaccharide; TLR4: Toll-like receptor 4; IL-6: Interleukin- 6; TNF-α: tumor necrosis factor; FGF-2: Fibroblast growth factor 2; Akt: Protein Kinase B; mTOR: mammalian target of rapamycin; S6K1: ribosome protein subunit 6 kinase 1; PAI-1: Plasminogen activator inhibitor-1; TF: Tissue factor.
Figure 2. Antibacterial mechanism of WIKE-14, PS1-2, and Pep19-2.5. TLR2: Toll-like receptor 2; TNF-α: tumor necrosis factor; TLR4: Toll-like receptor 4; LPS: Lipopolysaccharide; Pep19-2.5: As-pidasept; NF-κb: nuclear factor kappa-b.
Figure 3. Antioxidant mechanism of α-ch, FGF-19, MFG-E8, MG53, and GDF7. α-ch :Alpha-Chymotrypsin; FGF-19:Fibroblast growth factor 19; GDF7:Growth Differentiation Factor 7; MFG-E8 :As a biomarker of sepsis, Milk fat globule epidermal growth factor 8; MG53:Mitsugumin-53; TLR4:Toll-like receptor 4; NF-κb: nuclear factor kappa-b; GSH: Glutathione; SOD: Superoxide Dismutase; MPO: myeloperoxidase; iNOS: inducible nitric oxide synthase; MDA: malondialdehyde; NOx: nitrite/nitrate; LA: Linoleic acid; GLA: Gamma linolenic Acid; NRF2: Nuclear factor erythroid2-related factor 2; HO-1: Heme Oxygenase-1; ROS: Reactive oxygen species; Keap1: Recombinant Kelch Like ECH Associated Protein 1; GPX4: Glutathione peroxidase 4; PPARα: Peroxisome Proliferator-Activated Receptor-Alpha; bax: BCL2-Associated X; STING: Stimulator of interferon genes; AMPK: Adenosine 5‘-monophosphate (AMP)-activated protein kinase.
Figure 4. Anti-apoptotic mechanism of HGF, IL-7, IL-15, IL-22, Ac2-26, Vaspin, Hsp22, and APN. HGF: Hepatocyte growth factor ; IL-7:Interleukin-7; IL-15:Interleukin-15; IL-22:Interleukin-22; Vaspin: Serpin A12; Hsp22: Heat shock protein 22; APN: Adiponectin; c-Met: Cellu-lar-mesenchymal epithelial transition factor; PI3K: Phosphoinositide 3-Kinase; Akt: Protein Kinase B; mTOR: mammalian target of rapamycin; NF-κb: nuclear factor kappa-b; JAK: Just another kinase; STAT3/5: signal transducer and activator of transcription 3/5; bcl-2: B-cell lymphoma-2; IFN-γ: interferon-γ; PUMA: p53 up-regulated modulator of apoptosis; Bim: BCL-like 11; ATF4: Activating transcription factor 4; ATG7: Recombinant Autophagy Related Protein 7; S100A9: S100 calci-um-binding protein A9; bax: BCL2-Associated X; KLK7: Kallikrein 7; AMPK: Adenosine 5‘-monophosphate (AMP)-activated protein kinase; mTOR: mammalian target of rapamycin; ROS: Reactive oxygen species.
Figure 5. Anti-inflammatory mechanism of S100A8, CXT, Apelin-13, and SLPI. S1000A8: S100 calcium-binding protein A8; CTX: Crotoxin ; SLPI: Secretory leukocyte protease inhibitor; TLR4: Toll-like receptor 4; p38MAPK:P38 mitogen-activated protein kinases; IL-10: Interleukin-10; IL-6: Interleukin-6; TNF-α: tumor necrosis factor; LXA4: Lipoxin A4; PGE2: Prostaglandin E2; APJR: Apelin receptor; LPS: Lipopolysaccharide; TGF-β:Transforming growth factor betaï¼›SMAD2/3: drosophila mothers against decapentaplegic protein 2/3; NOX4: NADPH Oxidase 4; ROS: Reactive oxygen species; PFKFB3: 6-phosphofructo-2-kinase; IKK-α/β:inhibitor of kappa B kinase-α/βï¼›Iκb-α: inhibitor kappa B alphaï¼›NF-κb: nuclear factor kappa-b; MAPK: Mitogen-activated protein kinase.
Figure 6. Therapeutic mechanism of anti-HMGB1pAb, atezolizumab, secukinumab, adrecizumab, and tocilizumab. HMGB1: High mobility group box chromosomal protein 1; TLR4: Toll-like receptor 4; IRF3: Interferon regulatory factor 3; NF-κb: nuclear factor kappa-b; IL-4:Interleukin-4; IL-10:Interleukin-10; IL-6:Interleukin-6; TNF-α: tumor necrosis factor; PD-L1: Programmed cell death-Ligand 1; IκBα:Inhibitor kappa B alpha; S100A12: S100 calcium-binding protein A12; NLRP3: NOD-like receptor thermal protein domain associated protein 3; JNK: Jun N-terminal Kinase; IL-1β: Interleukin-1β.
Figure 7. Therapeutic mechanism of miR-25-5p, miR-214-3p, miR-142-5p, miR-340-5p, miR-26a-5p, and miR-490-3p. miR : microRNA TXNIP: Thioredoxin interacting protein Gene; NLRP3: Nucleo-tide- binding oligomerization domain; CTSB: cathepsin B; bax: BCL2-Associated X; ROS: reactive oxygen species; MDA: malondialdehyde; SOD: superoxide dismutase; bcl-2: B-cell lymphoma-2; NF-κb: nuclear factor kappa-b; MAPK: Mitogen-activated protein kinase; GSH: Glutathione; MyD88: Myeloid differentiation primary response gene 88; CTGF: Connective tissue growth factor; IL-6:Interleukin-6; TNF-α: tumor necrosis factor; IRAK1: interleukin 1 receptor associated kinase 1; TRAF6: TNF receptor associated factor 6.
Figure 8. Treatment mechanism of UFH, NAH, lentinan, LBP, and PCP. UFH :Heparin; NAH :N-heparin; LBP: Lycium barbarum polysaccharide; PCPs :Poria cocos polysaccharides; Xa: Factor Xa; Xâ… a: Factor Xâ… a; â… Xa: Factor â… Xa; Xâ… â… a: Factor Xâ… â… a; HMGB1: High mobility group box-1 protein; LPS: Lipopolysaccharide; GSDMD: Gasdermin D; HPA: heparinase; NF-κb: nuclear factor kappa-b; bax: BCL2-Associated X; TLR4: Toll-like receptor 4; MPO: myeloperoxidase; MDA: malondialdehyde; IL-6:Interleukin-6;TNF-α: tumor necrosis factor;IL-1β:Interleukin-1β; Nedd4: Neural precursor cell expressed developmentally down-regulated protein 4; Nedd4L: NEDD4 Like E3 Ubiquitin Protein Ligase; Gnb2: G protein subunit β2; PKM2: Pyruvate kinase 2; GSH: Glutathione; SOD: Superoxide Dismutase.
- The major limit in my opinion is that your paper provides a long list of macromolecules tested in sepsis, but it does not specify the real context (sepsis, septic shock?), the trial type (retospective study, randomized?), the sample size (humans, animals?) the effect (mortality? inflammation? how inflammation was detected?) the real clinical effect, the mechanism, and so on. Moreover, it does not critically discuss and consider all the limits of different molecules, with different actions, with different populations, with different evidences. You analyze vasopressors, anticoagulants, antibodies, and so on but they have different mechanism with different indications during sepsis and septic shock. No mention about this important concern.
Response:
We express our sincere gratitude for the valuable suggestion provided by the reviewer. The manuscript has been enhanced with additional information pertaining to the background, type of experiment, sample size, effect, and mechanism, all of which have been highlighted in red font. The details are as follows:
(1) Line 52-54
The inflammation detection methods described in these articles involve qPCR and ELISA technologies to measure the mRNA and protein levels of cytokines such as IL-1β, IL-6, and TNF-α among others.
(2) Line 64-84
C1-INH is a serine protease inhibitor that inhibits complement activation and re-duces coagulation in the blood. In a clinical study involving 61 patients with sepsis, high doses of C1-INH have been shown to reduce inflammatory responses in patients with sepsis and significantly decrease the 28-day mortality rate to below 12% [8].
Antithrombin (AT) has anticoagulant and anti-inflammatory effects. Recombi-nant antithrombin gamma (RAT-γ) is a recombinant form of AT lacking fucose. A ret-rospective study of 49 patients with sepsis with antithrombin deficiency and DIC found that the RAT-γ (36 IU/kg) and AT treatment groups showed a significant reduc-tion in JAAM-DIC and SOFA scores in sepsis patients. Additionally, the coagulation disorders and organ failure in these patients were improved [9].
Recombinant thrombomodulin (rhTM) can increase the expression of Glypcan1 and reduce the levels of inflammatory factors in the serum, improve cardiac microcir-culation disorder and vascular injury caused by sepsis, and enhance the 48-h survival rate of lipopolysaccharide (LPS)-induced sepsis mouse models. A retrospective study on patients with DIC revealed that rhTM treatment significantly reduced the DIC score, CRP levels, and SOFA score, improving coagulation function, inflammatory re-sponse, and organ damage [10,11].
Human soluble thrombomodulin (ART-123) is a recombinant soluble form of hu-man thrombomodulin. The multinational SCARLET phase III study included 800 pa-tients and reported that ART-123 effectively improved abnormalities in patients with sepsis-associated coagulation disease, but did not significantly improve the 28-day survival rate [12].
(3) Line 106-115
Polymyxin B (PMB) exerts both antibacterial and neutralizing functions. A pro-spective cohort study involving 60 patients with abdominal infectious sepsis found that PMB effectively reduced the level of endotoxin activity in septic shock patients and significantly improved their SOFA scores, but not survival [15]. Additionally, a meta-analysis of 12,234 critically ill adult patients showed that PMB has a higher probability of causing nephrotoxicity in patients [16].
The PS1-2 peptide has shown enhanced antifungal activity through membrane dissolution and inhibited TLR2 activity and TNF-a expression levels in RAW264. It ef-fectively combat sexually transmitted infections and inflammatory responses without triggering drug resistance [18].
(4) Line 119-130
WIKE-14, an antimicrobial peptide with antibacterial and anti-inflammatory ac-tivities, can neutralize endotoxins by inhibiting the binding of endotoxins to macro-phages, thereby inhibiting the expression of inflammatory factors, and reduce ROS and NO production induced by LPS. In the LPS-induced sepsis mouse model, WIKE-14 ef-fectively improved LPS-induced acute lung injury (ALI) [20].
Aspidasept (Pep19-2.5) is an anti-lipopolysaccharide polypeptide with a high binding affinity to LPS, neutralizes endotoxins, and exerts anti-inflammatory effects. In vitro cell and animal model toxicity tests demonstrate its safety since the therapeutic dose (30–50 μg/kg) is significantly lower than the toxic dose (2 mg/kg). Pep19-2.5 can reduce LPS-induced pyroptosis by preventing LPS from binding to TLR4, inhibiting the activation of NF-κB pathway, reducing the inflammatory response caused by LPS, and inhibiting the activation of caspase-1[21,22].
(5) Line 140-162
Alpha-Chymotrypsin (α-ch) is a serine protease. By inhibiting the TLR4/NF-κB pathway, α-ch, a serine protease, reduces the levels of inflammatory cytokines IL-1β, IL-6, and TNF-α and the expression of MPO and iNOS in a cecum ligation and punc-ture (CLP)-induced sepsis mouse model. After α-ch treatment, the survival rate of CLP mice increased by 50% [25].
Fibroblast growth factor 19 (FGF-19) can reduce the expression of LA and GLA, activate the NRF2/HO-1 pathway, and inhibit the expression of INOS, ROS, caspase-3, and cytochrome c, thereby alleviating lipid metabolism disorders, liver injury, and kidney injury in LPS-induced sepsis mice [26].
Growth Differentiation Factor 7 (GDF7) can reduce inflammation and oxidative stress by downregulating STING-activated AMPK signaling, improving LPS-induced lung injury in septic mice; the 72-h survival rate of septic mice treated with GDF7 ex-ceeds 50% [27].
As a biomarker of sepsis, milk fat globule epidermal growth factor 8 (MFG-E8) significantly reduces the levels of inflammatory factors and the concentration of MDA and ferrous ions in the liver, increased the ratio of GSH and GPX4 in the antioxidant system, and inhibits oxidative stress and ferroptosis in the liver of CLP mouse model. MFG-E8 has a protective effect on liver injury and increases the survival rate of CLP mice by >40% [28].
Mitsugumin-53 (MG53) is a protein of the TRIM family that exhibits an-ti-inflammatory and anti-oxidative properties, promotes plasma membrane repair, and improves myocardial dysfunction caused by sepsis by upregulating PPARα ex-pression. Additionally, treatment with MG53 has been shown to increase the 72-h sur-vival rate (>60%) of CLP rats [29].
(6) Line 180-192
Hepatocyte growth factor (HGF) activates the mTOR pathway through target c-Met, reduces apoptosis in liver and lung tissue, and reduces the levels of IL-1 β, IL-18, LDH, and ROS, and alleviates the inflammatory and oxidative stress induced by sepsis. HGF treatment enhanced the survival rate of CLP mice to over 50% [30]. Interleukin-7 (IL-7) promotes the secretion of Bcl-2 and IFN- γ through the JAK/STAT pathway, in-hibits the expression of pro-apoptotic protein Bim. A preclinical trial involving 70 pa-tients with septic shock demonstrated that IL-7 restored the levels of CD4+ and CD8+ T cells in sepsis patients and improved the lymphocyte dysfunction caused by sepsis [31]. Interleukin-15 (IL-15), a cytokine belonging to the IL-2 family, can increase the ratio of T cells and natural killer (NK) cells in septic rats by promoting the expression of Bcl-2 and IFN- γ and inhibiting the expression of pro-apoptotic protein Bim. After IL-15 injection, the survival rates of CLP mice and mice infected with Pseudomonas ae-ruginosa increased three- and two-fold, respectively [32]. .
(7) Line 198-205
IL-38 is a cytokine of the IL-1 family that inhibits the NLRP3/IL-1β signaling pathway and Treg cell immune function, reduces the expression of inflammatory me-diators and pro-apoptotic proteins, attenuates early inflammatory response and apop-tosis, improves the depletion of effector T cells during sepsis, and increases the surviv-al rate of septic mice (>60%) [34,35].
Fibroblast growth factor 15 (FGF-15) is an analog of FGF-19 FGF-15 that signifi-cantly improves liver inflammation and apoptosis in septic mice by targeting FGFR4, reducing the proportion of Tregs, and increasing the survival rate of septic mice by 40% [36]. .
(8) Line 212-221
Ac2-26 is an N-terminal active peptide of AnnexinA1 that targets Fpr2 receptors and inhibits PI3K and AKT phosphorylation. Ac2-26 downregulates the level of NF-κB, reduces the expression of inflammatory cytokines and pro-apoptotic proteins in septic mice, inhibits inflammation and apoptosis, improves renal injury, and increases the 7-day survival rate of CLP mice (>40%) [40].
Vaspin (Serpin A12) is an adipose factor of the Serpin family, which can inhibit the expression of KLK7, down-regulate the expression of pro-apoptotic protein bax and TNF-α and other inflammatory factors, promote the expression of anti-apoptotic pro-tein bcl-2, reduce inflammatory response and apoptosis, and improve septic heart in-jury in mice. Vaspin treatment increased the survival rate of CLP mice by 50% [41].
(9) Line 250-263
HSA regulates the blood osmotic pressure [85] and improves edema, hypoalbu-minemia, and hypovolemia caused by sepsis. A retrospective study involving 5,009 pa-tients with sepsis demonstrated that treatment with 5% HSA significantly reduced sepsis-related 28-day mortality (8.7%) [45].
Selepressin is a selective vasopressin V1a agonist. A phase IIa randomized, place-bo-controlled trial with 53 patients with septic shock showed that Selepressin (2.5 ng/kg/min) could replace norepinephrine (18.24 μg/kg/min) to maintain mean arterial pressure balance with good tolerance [46]. However, a phase III/IIb clinical study with 868 patients with septic shock revealed that Selepressin did not significantly reduce the usage of ventilators and pressor drugs in patients with septic shock within 30 days [47].
B38-CAP, an ACE2-like enzyme derived from Bacillus-like bacteria, can upregu-late the level of angiotensin 1–7, reduce the levels of AngII and inflammatory cyto-kines and chemokines, improve lung injury caused by sepsis, and increase the survival rate of CLP mice by 20% [48]. .
(10) Line 272-276
Erythropoietin (EPO) is a glycoprotein with a molecular weight 30kDa. A retro-spective study involving 344 patients with sepsis demonstrated that high-dose EPO (8,000–16,000 units/week) improved the 29-day survival rate (85%), without increas-ing the incidence of adverse events such as thrombosis. Furthermore, high-dose EPO reduced mortality and inflammation in a septic mouse model [50]. .
(11) Line 281-287
Crotoxin (CTX) is a protein derived from the South American rattlesnake venom. In the model of CLP sepsis, CTX significantly downregulated the expression of pro-inflammatory cytokines TNF-α and IL-6, and increased the ratio of LXA4, PGE2, and IFN-γ in mice plasma. In the bone marrow derived macrophage of mice with sep-sis, CTX could improve the inflammatory response caused by sepsis and reduce the mortality of CLP mice by regulating phagocytosis and promoting the secretion of ROS and NO, and reduced the mortality of CLP mice by 40% [52]. .
(12) Line 308-312
AB103 is a CD28 antagonist that inhibits inflammation and improves the efficiency of bacterial clearance in a bacterial sepsis model. A multicenter, randomized clinical trial involving 40 patients with NSTI showed that AB103 could improve organ damage and safety. However, it did not significantly affect the duration of ventilator use or the level of cytokines [23,24].
(13) Line 330-360
IVIG is a mixed IgG antibody. A single-center retrospective study of 239 patients with sepsis showed that IVIG significantly improved the prognosis of patients with low IgG levels and reduced mortality. A retrospective multicenter study of 850 pa-tients with severe COVID-19 showed that IVIG did not significantly improve the 28-day survival rate of patients, and might have induced adverse thrombosis and al-lergic reactions [58,59].
HMGB1 is an inflammatory mediator involved in sepsis. Anti-HMGB1 antibody is a monoclonal antibody developed against the inflammatory mediator HMGB1 that can regulate the levels of pro-inflammatory and anti-inflammatory factors, improve the inflammatory response in CLP mice, restore hemoglobin and hematocrit, improve anemia, enhance anti-infection ability, and reduce the mortality rate of CLP mice by 50% [60,61].
Atezolizumab is a PD-L1 monoclonal antibody, which can treat sepsis by blocking the PD-1/PD-L1 pathway. In sepsis, atezolizumab can reduce the level of endotoxins in the CLP model, increase the expression levels of claudin-1 and occludin proteins, im-prove intestinal barrier function, reduce T cell apoptosis, and alleviate immunosup-pression , and reduce the mortality of CLP mice by 25% within 90 h of treatment [62].
Vilobelimab is a monoclonal antibody against the allergic toxin C5a that amelio-rates infectious organ dysfunction by neutralizing C5a. A randomized, double-blind, multicenter phase IIa study of 72 patients with sepsis or septic shock showed that vi-lobelimab neutralizes C5a in a dose-dependent manner with good tolerance and safety. However, the study did not show significant improvement in sepsis prognosis and mortality [63].
Secukinumab is a human monoclonal antibody that neutralizes IL-17A to inhibit the IKB/NF-κB pathway, reduce the expression of pro-inflammatory factors, such as IL-6 and TNF-α, and improve the inflammatory response and lung injury induced by sepsis. Simultaneously, the survival rate of CLP rats injected with a high-dose secuki-numab (20 mg/kg) was 60% after 196 h [64].
Adrecizumab (HAM8101) is a non-neutralizing human adrenomedullin mono-clonal antibody that reduces the expression of VEGF, increases the level of Angiopoi-etin-1, improves the effect of sepsis on vascular permeability, and reduces inflamma-tion and mortality in septic mice [65]. . A phase 2, double-blind, randomized, con-trolled trial involving 301 patients with septic shock found that adrecizumab was well tolerated but did not significantly improve the survival rate of patients [88].
(14) Line 379-403
miR-25-5p can improve brain injury caused by sepsis by inhibiting the TXNIP/NLRP3 pathway; reducing the levels of TXNIP, NLRP3, and cleaved caspase-1; and inhibiting the expression of inflammatory factors (such as IL-6, IL-1β, and TNF-α) and peroxide, reduce LPS-induced inflammatory and oxidative stress and apoptosis, and improve LPS-induced brain injury in septic rats [67]. miR-214-3p can inhibit the expression of CTSB, regulate the levels of anti-apoptotic bcl-2 and pro-apoptotic pro-teins Bax and caspase-3, decrease the levels of ROS and MDA, increase the proportion of SOD, reduce the apoptosis and oxidative damage of AC16 cells stimulated by LPS, and improve myocardial injury caused by sepsis [68]. miR-340-5p can reduce the ex-pression of MyD88 protein, inhibit the elevated production of ROS and MDA induced by LPS, increase the level of GSH, reduce the oxidative stress response of septic SIC cell model, and improve LPS-induced HL-1 cell injury. miR-340-5p can reduce the degree of myocardial injury and oxidative stress in LPS sepsis-induced cardiomyopathy model [69]. miR-26a-5p can inhibit the expression of inflammatory factors (TNF-α, IL-1β, and IL-6) and pro-apoptotic protein Bax, reduce pulmonary inflammation and apoptosis in mice, and improve pulmonary inflammation and apoptosis, and increase the survival rate of septic ALI mice by 30% [70]. By targeting IRAK1, miR-490-3p inhibits the IRAK1/TRAF6 pathway, reduces the level of NF-κB phosphorylation; decreases the ex-pression of inflammatory factors IL-1β, IL-6, and TNF α in ALI rat model; and im-proves LPS-induced pulmonary inflammation and apoptosis in rats [71]. Furthermore, the inhibition of the NF-κB pathway through the targeting of IRAK1/TRAF6 by miR-146a has been observed to effectively reduce the inflammatory response in mice subjected to CLP, as well as mitigate splenocyte apoptosis. Notably, the administration of miR-146a resulted in a significant improvement in sepsis-induced organ damage, leading to a noteworthy 40% increase in the survival rate of CLP mice (n=15) [72].
(15) Line 423-430
UFH, a glycosaminoglycan, and NAH, and non-anticoagulant, can reduce lung leukocyte infiltration, capillary barrier injury, and pulmonary edema in septic mice, inhibit caspase-11-induced cell pyrolysis induced by LPS and HMGB1, reduce the increase of pro-inflammatory cytokines IL-1 β and TNF-α, and inhibit HPA activity to protect endothelial glycocalyx. The anti-inflammatory effect of UFH is superior to that of NAH; however, UFH increases the risk of bleeding. A retrospective study of 3,377 SIC patients with sepsis found that UFH (6250-13750 IU/d) reduced the risk of death in patients with an SIC score of 4; however, UFH increased the risk of bleeding in clinical trial [73,91,92]
(16) Line 442-460
PCPs is a polysaccharide component of the traditional Chinese medicine Poria co-cos, with various biological activities, such as anti-inflammation, antioxidation, and immune regulation. In a septic mice model, PCP preconditioning significantly de-creased the expression of inflammatory cytokines IL-6, IL-1 β, and TNF-α, improved inflammation and oxidative stress in FP mice by downregulating the levels of MDA and MPO. PCP can improve the inflammation and oxidative stress response of FP mice, reduce splenocyte apoptosis, regulate the proportion of spleen Tregs, and increase the survival rate of sepsis mouse models by approximately 30% [76].
Tannic acid is a polyphenolic compound with both anti-inflammatory and anxio-lytic properties. In sepsis, tannic acid can regulate blood pressure, reduce the expres-sion of TNF-α, IL-6, IL-1 β and MDA in brain tissue, upregulate the expression of GABAA in the hippocampus, reduce inflammation and oxidative stress in patients with sepsis, decrease inflammation and oxidative stress in CLP rats; and improve anx-iety-related behaviors in SAE rats [77].
Eritoran is a derivative of lipopolysaccharide (LPS) lipid A and a TLR4 blocker. Eritoran can inhibit the expression of inflammatory mediators, inhibit the cytokine storm in sepsis, increase Th1 reaction time, and reduce death in mice with bacterial sepsis. However, in a randomized, double-blind, phase 3 clinical trial of 1,961 subjects, eritoran could not reduce the 28-day mortality in patients with severe sepsis [78,93].

Round 2
Reviewer 3 Report
Authors extensively revised the manuscripts.
I appreciate their efforts.
Author Response
August 4, 2023
Prof. Dr. Maurizio Battino
Editor-in-Chief
Ms. Maria Otilia Drugan
Assistant Editor
International Journal of Molecular Sciences
Dear Editor,
I wish to re-submit the manuscript titled “Research progress of macromolecules in the prevention and treatment of sepsis.” The manuscript ID is ijms-2513678.
We would like to extend our sincere appreciation to both you and the reviewers for the invaluable suggestions and insightful perspectives provided. The manuscript has greatly benefited from these astute recommendations.
Attached is the revised version of our manuscript. In the following pages are our point-by-point responses to each of the comments of the reviewers. Revisions in the text are highlighted by the utilization of the color red. We hope that the revisions in the manuscript and our accompanying responses would be sufficient to make our manuscript suitable for publication in International Journal of Molecular Sciences.
Thank you for your consideration. I look forward to hearing from you.
Sincerely,
Jingqian Su, Ph.D.
Associate Professor
Fujian Key Laboratory of Innate Immune Biology
Biomedical Research Center of South China
College of Life Science, Fujian Normal University
Fuzhou 350117, Fujian, China
Tel: +86-18950498937
E-mail: sjq027@fjnu.edu.cn
Responses to the comments of Reviewer #3
- Authors extensively revised the manuscripts. I appreciate their efforts.
Respond:
We express our sincere gratitude to the reviewer for the invaluable suggestions.

Reviewer 4 Report
The papaer is improve, but I still found difficulties in reading.
Please specify RAW264. and also MDA line 155., and also CLP rats.
Please specify JAK/STAT pathway, inhibits the 185 expression of pro-apoptotic protein Bim, line 185, Tregs line 206.
What is septic SE? and SAE?
And so on.
All abbreviations must be specified in text, no only in figures. The same applies to table 1, for which I do not see any legend explainig all abbreviations.
Moreover, antibiotica are recommended for sepsis management, not only used as you stated.
Considering polimyxin, do you refer to extracorporeal techniques? Please specify. Moreover, these methods are no longerv recommended.
In the paragraph 2.5. Regulators of blood pressure and volume, you speak about albumion without citing a body of literature on albumin and sepsis in terms of glycocalix protection, vascular effect, volume restoring, and so on.
You cite selepressin , and what about vasopressin that has much more supporting papers?
Several mechanism of action of macromolecules need to be better clarified for the reader (see for example line 388" miR-340-5p can reduce the expression of MyD88 protein, inhibit the elevated production of ROS and MDA induced by LPS, increase the level of GSH, reduce the oxidative stress response of septic SIC cell model, and improve LPS-induced HL-1 cell injury." specify My D88 and Sic. )
In discussion, you should underline the limits of different molecules, the limited evidence in most cases to 1 human study or to preclinical ones, the lack of large human studies. Do you know if there are other ongoing trials on the macromolecules? If no, why? Maybe other evidences would clarify the effects of macromolecules.
English needs only minor revisions.
Author Response
August 4, 2023
Prof. Dr. Maurizio Battino
Editor-in-Chief
Ms. Maria Otilia Drugan
Assistant Editor
International Journal of Molecular Sciences
Dear Editor,
I wish to re-submit the manuscript titled “Research progress of macromolecules in the prevention and treatment of sepsis.” The manuscript ID is ijms-2513678.
We would like to extend our sincere appreciation to both you and the reviewers for the invaluable suggestions and insightful perspectives provided. The manuscript has greatly benefited from these astute recommendations.
Attached is the revised version of our manuscript. In the following pages are our point-by-point responses to each of the comments of the reviewers. Revisions in the text are highlighted by the utilization of the color red. We hope that the revisions in the manuscript and our accompanying responses would be sufficient to make our manuscript suitable for publication in International Journal of Molecular Sciences.
Thank you for your consideration. I look forward to hearing from you.
Sincerely,
Jingqian Su, Ph.D.
Associate Professor
Fujian Key Laboratory of Innate Immune Biology
Biomedical Research Center of South China
College of Life Science, Fujian Normal University
Fuzhou 350117, Fujian, China
Tel: +86-18950498937
E-mail: sjq027@fjnu.edu.cn
Responses to the comments of Reviewer #4
1.Please specify RAW264. and also MDA line 155., and also CLP rats.
Respond:
We would like to extend our heartfelt appreciation for the invaluable suggestions offered by the reviewer, while also expressing deep regret for the error that was made on our part. RAW264 is a typo and the full names of RAW264.7cells, MDA and CLP have been added to the text in the manuscript,as follows:
(1) Line 136
TLR2 activity and TNF-α expression levels in RAW264.7 cells.
(2) Line 180
malondialdehyde (MDA)
2.Please specify JAK/STAT pathway, inhibits the 185 expression of pro-apoptotic protein Bim, line 185, Tregs line 206.What is septic SE? and SAE?
Respond:
Many thanks to the reviewer for their valuable suggestions. We provide a comprehensive elucidation of the JAK/STAT pathway, while also presenting a detailed explanation of septic encephalopathy (SE) and sepsis-associated encephalopathy (SAE) as SE and SAE, respectively; as follows:
(1) Line 214-217
IL-7 promotes the secretion of B-cell lymphoma-2 (Bcl-2) and interferon-γ (IFN- γ) through an activated kinase (JAK), promotes signal transducer and activator of transcription (STAT) phosphorylation, and inhibits the expression of pro-apoptotic protein BCL-like 11(Bim)
(2) Line 242
septic encephalopathy (SE)
(3) Line 245
sepsis-associated encephalopathy (SAE)
3.All abbreviations must be specified in text, no only in figures. The same applies to table 1, for which I do not see any legend explainig all abbreviations.
Response:
We would like to extend our heartfelt appreciation for the invaluable suggestion put forth by the reviewer. The complete name has been indicated with its corresponding abbreviation upon its initial mention in both the text and table 1, as follows:
(1) Line 60-77
ADM: Adrenomedullin; AKT: protein kinase B; AMPK: adenosine 5‘-monophosphate (AMP)-activated protein kinase; AT: Antithrombin; ATF4: activating transcription factor 4; ATG7: recombinant autophagy related protein 7; C5a: component 5a; c-Met: cellular mesenchymal–epithelial transition factor; CTGF: connective tissue growth factor; CTSB: cathepsin B; FGFR4: fi-broblast growth factor receptor 4; Fpr2: Formyl-peptide receptor-2; GABAA: γ-aminobutyric acid sub-type A; GPX4: glutathione peroxidase 4; GSDMD: Gasdermin D; HMGB1: high-mobility group protein 1; IL10R: interleukin-10 receptor; IL-15R: interleukin-15 receptor; IL-1β: interleukin-1β; IL-22R: interleukin-22 receptor; IRAK1: interleukin 1 receptor associated kinase 1; JAK: Just another kinase; KLK7: kallikrein 7; LPS: lipopolysaccharide; MAP: mean arterial pressure; MAPK: mito-gen-activated protein kinase; MITOL: mitochondrial ubiquitin ligase; mTOR: mammalian target of rapamycin; MyD88: myeloid differentiation factor 88; NADPH oxidase; ROS: reactive oxygen species; NF-κB: nuclear factor kappa-light-chain-enhancer of activated B cells; NLRP3: NOD-like receptor thermal protein domain associated protein 3; NRF2: nuclear factor erythroid 2-related factor 2; HO-1: heme oxygenase-1; PD-L1: programmed cell death-ligand 1; PFKFB3: 6-phosphofructo-2-kinase; PI3K: phosphoinositide 3-kinase; PKM2: pyruvate kinase 2; S6K1: ri-bosome protein subunit 6 kinase 1; SMAD: drosophila mothers against decapentaplegic protein; STAT: signal transducer and activator of transcription; STING: stimulator of interferon genes; TATc: thrombin–antithrombin complex; TGF-β1: Transforming growth factor β1; TLR4: Toll-like receptor 4; TXNIP: thioredoxin interacting protein;
(2) Line 81-82
Direct thrombin inhibitors and factor Xa inhibitors
(3) Line 83-84
Fibroblast growth factor 2 (FGF-2)
(4) Line 85
C1-esterase inhibitor (C1-INH)
(5) Line 92-93
Japanese Association for Acute Medicine-DIC
(6) Line 97-100
lipopolysaccharide (LPS)-induced sepsis mouse models. A retrospective study on pa-tients with DIC revealed that rhTM treatment significantly reduced the DIC score, C-reactive protein levels
(7) Line 107-109
the protein kinase B (AKT)/ mammalian target of rapamycin (mTOR)/ ribosome pro-tein subunit 6 kinase 1 (S6K1) phosphorylation. Moreover, it reduces the expression levels of plasminogen activator inhibitor-1 and tissue factor
(8) Line 111-113
mitogen-activated protein kinase (MAPK) activation and reduces inflammation by blocking the binding of LPS and high-mobility group protein 1 (HMGB1) to the toll-like receptor 4 (TLR4)/myeloid differentiation protein-2 complex
(9) Line 130-131
polymyxin B-hemoperfusion (PMX-HP)
(10) Line 144-145
reactive oxygen species (ROS) and nitric oxide (NO) production
(11) Line 152
nuclear factor kappa-light-chain-enhancer of activated B cells (NF-Κb)
(12) Line 167-169
and inhibits the expression levels of myeloperoxidase (MPO) and inducible nitric ox-ide synthase (iNOS) in a cecum ligation and puncture (CLP)-induced sepsis mouse model
(13) Line 170-171
linoleic acid (LA) and gamma linolenic acid, activate the nuclear factor erythroid 2-related factor 2/heme oxygenase-1 (HO-1) pathway
(14) Line 175-177
stimulator of interferon genes, promoting adenosine 5‘-monophosphate (AMP)-activated protein kinase (AMPK) phosphorylation
(15) Line 180-181
malondialdehyde (MDA) and ferrous ions in the liver, increases the ratio of glutathi-one (GSH) and glutathione peroxidase 4
(16) Line 187-188
proliferator activated receptor-alpha
(17) Line 209-212
HGF inhibits phosphoinositide 3-kinase (PI3K) and AKT phosphorylation acti-vates the mTOR pathway,and inhibits the activation of NF-κB through target cellular mesenchymal–epithelial transition factor; reduces apoptosis in liver and lung tissue; and reduces the levels of IL-1 β, IL-18, lactated hydrogenase (LDH)
(18) Line 214-217
IL-7 promotes the secretion of B-cell lymphoma-2 (Bcl-2) and interferon-γ (IFN- γ) through an activated kinase (JAK), promotes signal transducer and activator of tran-scription (STAT) phosphorylation, and inhibits the expression of pro-apoptotic protein BCL-like 11(Bim)
(19) Line 226-229
promoting the expression of S100 calcium-binding protein A9 (S100A9) in the liver, reducing inflammation, and activating the STAT3 and activating transcription factor 4, which are recombinant autophagy related protein 7
(20) Line 231-233
NOD-like receptor thermal protein domain associated protein 3 (NLRP3)/IL-1β sig-naling pathway, inhibits the activation of NLRP3 inflammasomes, enhances CD4CD25 Treg cell immune function
(21) Line 238
fibroblast growth factor receptor 4
(22) Line 242
septic encephalopathy (SE)
(23) Line 245
sepsis-associated encephalopathy (SAE)
(24) Line 246-247
formyl peptide receptor-2 receptors and inhibits PI3K and AKT phosphorylation
(25) Line 252-253
kallikrein 7; downregulate the expression of pro-apoptotic protein BCL2-associated X (bax)
(26) Line 267-270
fibronectin type III domain-containing protein 5. Irisin can decrease the levels of IL-1 β and Gasdermin D (GSDMD) by regulating the mitochondrial ubiquitin ligase (MI-TOL)/GSDMD pathway, improving the levels of LDH and creatine kinase-MB
(27) Line 313-314
alanine aminotransferase, alkaline phosphatase, and aspartate aminotransferase
(28) Line 321
S100 calcium-binding protein A8 (S100A8)
(29) Line 328
lipoxin A4, prostaglandin E2
(30) Line 335
macrophage inflammatory protein 1-a and chemotokine ligand 5
(31) Line 338-339
myeloid-derived suppressor cells
(32) Line 345
NADPH oxidase 4
(33) Line 348-349
transforming growth factor beta, drosophila mothers against decapentaplegic protein (SMAD) 2, and SMAD 3
(34) Line 352
monocyte chemoattractant protein-1
(35) Line 356
necrotizing soft-tissue infections
(36) Line 377
IVIG (intravenous immunoglobulins)
(37) Line 388-389
programmed cell death-ligand 1 (PD-L1) monoclonal antibody, which can treat sepsis by blocking the programmed cell death protein 1 (PD-1)
(38) Line 394-395
complement component 5a (C5a)
(39) Line 400-402
Secukinumab is a human monoclonal antibody that neutralizes IL-17A to inhibit the inhibitor kappa B alpha (IκBα) /NF-κB pathway, reduce IκBα and NF-κB phos-phorylation levels, inhibit the expression of pro-inflammatory factors
(40) Line 407
vascular endothelial growth factor
(41) Line 414
superoxide dismutase (SOD)
(42) Line 435
cathepsin B
(43) Line 439
myeloid differentiation factor 88 protein
(44) Line 441
sepsis-induced cardiomyopathy (SIC)
(45) Line 447-448
By reducing the expression levels of IL-1 receptor associated kinase 1 (IRAK1), miR-490-3p inhibits the IRAK1/ TNF receptor associated factor 6 (TRAF6) pathway
(46) Line 479
Heparinase
(47) Line 490-493
LBP promotes pyruvate kinase 2 (PKM2) ubiquitination by increasing the expression of NEDD4 Like E3 ubiquitin protein ligase (Nedd4L), neural precursor cell expressed de-velopmentally downregulated protein 4 (Nedd4), and G-protein subunit β2 in Raw264.7 cells
(48) Line 502
fecal-induced peritonitis mice
(49) Line 507-508
γ- aminobutyric acid sub-type A
- Moreover, antibiotica are recommended for sepsis management, not only used as you stated.
Response:
We express our sincere gratitude to the reviewers for their insightful inquiries. As our manuscript focuses on the utilization of macromolecules in the context of sepsis, we have exclusively chosen macromolecular antibacterial agents, including antimicrobial peptides (line 125-128), for investigation.
- Considering polimyxin, do you refer to extracorporeal techniques? Please specify. Moreover, these methods are no longerv recommended.
Response:
We express our gratitude to the reviewer for the invaluable comments, which have greatly assisted us. More recently, extracorporeal techniques, such as the utilization of polymyxin B hemoperfusion therapy, have been employed in treating sepsis (line 129-133), as follows:
Polymyxin B (PMB) exerts both antibacterial and neutralizing functions. A prospective cohort study involving 60 patients with abdominal infectious sepsis found that polymyxin B-hemoperfusion (PMX-HP) treatment effectively reduced the level of endotoxin activity in patients with septic shock, but not improved their SOFA scores and survival rates
- In the paragraph 2.5. Regulators of blood pressure and volume, you speak about albumion without citing a body of literature on albumin and sepsis in terms of glycocalix protection, vascular effect, volume restoring, and so on.
Response:
We would like to extend our heartfelt appreciation for the invaluable suggestion provided by the reviewer. This section has been supplemented in line 290-293, as follow:
In addition, albumin can also reduce heme-induced hepatic sinusoidal contraction and cytotoxicity, improve liver microcirculation and organ function by binding heme; Albumin reduces pulmonary edema and improves endothelial dysfunction of the lungs [88,89].
- You cite selepressin, and what about vasopressin that has much more supporting papers?
Response:
We express our gratitude for your valuable comment and extend our sincere apologies for any confusion that may have arisen. This section has been supplemented in line 304-306, as follow:
VP can increase the mean arterial pressure, improve body perfusion, and reduce the number of renal replacement therapy in patients with septic shock in clinical prac-tice, but there is no significant improvement in patient mortality and ICU length of stay [79,80].
- Several mechanism of action of macromolecules need to be better clarified for the reader (see for example line 388" miR-340-5p can reduce the expression of MyD88 protein, inhibit the elevated production of ROS and MDA induced by LPS, increase the level of GSH, reduce the oxidative stress response of septic SIC cell model, and improve LPS-induced HL-1 cell injury." specify My D88 and Sic. )
Response:
We would like to extend our heartfelt appreciation for the invaluable suggestion put forth by the reviewer. In accordance with the recommendations provided by the reviewer, we have implemented comprehensive revisions to the complete manuscript, which are outlined as follows:
(1) Line 107-109
the protein kinase B (AKT)/ mammalian target of rapamycin (mTOR)/ ribosome pro-tein subunit 6 kinase 1 (S6K1) phosphorylation. Moreover, it reduces the expression levels of plasminogen activator inhibitor-1 and tissue factor,
(2) Line 175-177
GDF7 can reduce inflammation and oxidative stress by downregulating stimula-tor of interferon genes, promoting adenosine 5‘-monophosphate (AMP)-activated pro-tein kinase (AMPK) phosphorylation
(3) Line 209-212
HGF inhibits phosphoinositide 3-kinase (PI3K) and AKT phosphorylation acti-vates the mTOR pathway,and inhibits the activation of NF-κB through target cellular mesenchymal–epithelial transition factor; reduces apoptosis in liver and lung tissue; and reduces the levels of IL-1 β, IL-18, lactated hydrogenase (LDH)
(4) Line 214-217
IL-7 promotes the secretion of B-cell lymphoma-2 (Bcl-2) and interferon-γ (IFN- γ) through an activated kinase (JAK), promotes signal transducer and activator of tran-scription (STAT) phosphorylation, and inhibits the expression of pro-apoptotic protein BCL-like 11(Bim)
(5) Line 231-233
IL-38 is a cytokine of the IL-1 family that regulates the NOD-like receptor thermal protein domain associated protein 3 (NLRP3)/IL-1β signaling pathway, inhibits the ac-tivation of NLRP3 inflammasomes, enhances CD4CD25 Treg cell immune function
(6) Line 245-247
Ac2-26 is an N-terminal active peptide of Annexin A1 that targets formyl peptide re-ceptor-2 receptors and inhibits PI3K and AKT phosphorylation.
(7) Line 258-261
Hsp22 may inhibit the expression of apoptotic proteins and inflammatory factors by promoting AMPK activation and reducing mTOR phosphorylation levels, and signifi-cantly improves myocardial injury and inflammation in an LPS-induced sepsis mouse model
(8) Line 400-402
Secukinumab is a human monoclonal antibody that neutralizes IL-17A to inhibit the inhibitor kappa B alpha (IκBα) /NF-κB pathway, reduce IκBα and NF-κB phos-phorylation levels, inhibit the expression of pro-inflammatory factors
- In discussion, you should underline the limits of different molecules, the limited evidence in most cases to 1 human study or to preclinical ones, the lack of large human studies. Do you know if there are other ongoing trials on the macromolecules? If no, why? Maybe other evidences would clarify the effects of macromolecules
Response:
We express our gratitude for your valuable comment and extend our sincere apologies for any confusion that may have arisen. The paragraph (line 527-529) has been revised as follow:
The efficacy, dosage, side effects, and therapeutic mechanisms of biological macromolecules remain unclear and only a few have entered the clinical stage. Moreover, clinical data are limited, and larger clinical trials are needed to clarify these.

Round 3
Reviewer 4 Report
The paper is now very clear and I wish to congratulate with all authors for their impressive work.
Just check line s 203 and later for english (not so clear ) and do double check for all abbreviations (HSA for example.)
The paper is much improved, it needs only minor english checks.
Author Response
August 10, 2023
Prof. Dr. Maurizio Battino
Editor-in-Chief
Ms. Maria Otilia Drugan
Assistant Editor
International Journal of Molecular Sciences
Dear Editor,
I wish to re-submit the manuscript titled “Research progress of macromolecules in the prevention and treatment of sepsis.” The manuscript ID is ijms-2513678.
Please accept our sincere appreciation for the valuable suggestions and insightful perspectives provided by you and the reviewers. These astute recommendations have greatly enhanced the manuscript.
Attached is the revised version of our manuscript. In the following pages are our point-by-point responses to each of the comments of the reviewers. Revisions in the text are highlighted by the utilization of the color red. We hope that the revisions in the manuscript and our accompanying responses would be sufficient to make our manuscript suitable for publication in International Journal of Molecular Sciences.
Thank you for your consideration. I look forward to hearing from you.
Sincerely,
Jingqian Su, Ph.D.
Associate Professor
Fujian Key Laboratory of Innate Immune Biology
Biomedical Research Center of South China
College of Life Science, Fujian Normal University
Fuzhou 350117, Fujian, China
Tel: +86-18950498937
E-mail: sjq027@fjnu.edu.cn
Responses to the comments of Reviewer #1
- The paper is now very clear and I wish to congratulate with all authors for their impressive work.
Respond:
We express our sincere gratitude to the reviewer for the invaluable suggestions.
- Just check lines 203 and later for english (not so clear) and do double check for all abbreviations (HSA for example.)
Respond:
We would like to extend our heartfelt appreciation for the invaluable suggestions offered by the reviewer, while also expressing deep regret for the error that was made on our part.
(1) We have revised section 2.4 (Lines 203-269) based on the reviewer's comments. The specific modifications are described below:
2.4. Anti-apoptosis
Anti-apoptosis drugs inhibit apoptosis by regulating the ratio of anti-apoptotic proteins (Bcl-2 and Mcl1) to pro-apoptotic proteins (Bax and Bad) as well as inflammation and oxidative stress [83]. As shown in Figure 4, hepatocyte growth factor (HGF), IL-7, IL-15, IL-22, Ac2-26, Vaspin, Hsp22, and adiponectin (APN) exhibit anti-apoptotic functions in the prevention and treatment of sepsis.
HGF inhibits phosphoinositide 3-kinase (PI3K) and AKT phosphorylation, activates the mTOR pathway, and inhibits the activation of NF-κB through target cellular mesenchymal–epithelial transition factor; reduces apoptosis in the liver and lung tissue; reduces the IL-1 β, IL-18, lactated hydrogenase (LDH), and ROS levels; and alleviates sepsis-induced inflammatory and oxidative stress. HGF treatment has been shown to improve the survival rate of CLP mice to over 50% [30]. IL-7 promotes the secretion of B-cell lymphoma-2 (Bcl-2) and interferon-γ (IFN- γ) through an activated kinase (JAK), promotes signal transducer and activator of transcription (STAT) phosphorylation, and inhibits the expression of pro-apoptotic protein Bcl-like 11 (Bim).A preclinical trial including 70 patients with septic shock demonstrated that IL-7 restored the CD4+ and CD8+ T cell levels in patients with sepsis and improved the lymphocyte dysfunction caused by sepsis [31]. IL-15, a cytokine belonging to the IL-2 family, can increase the ratio of T cells and natural killer cells in septic rats by promoting the expression of Bcl-2 and IFN- γ and inhibiting the expression of pro-apoptotic protein, Bim. After IL-15 injection, the survival rates of CLP mice and mice infected with Pseudomonas aeruginosa increased by three- and two-fold, respectively [32].
IL-22, a member of the IL-10 family of cytokines, can induce the differentiation and development of inhibitory M2 macrophages by promoting the expression of S100 calcium-binding protein A9 (S100A9) in the liver, reducing inflammation, and activating STAT3 and transcription factor 4, which are recombinant autophagy related protein 7 hepatocyte autophagy signaling pathways to reduce liver inflammation and injury [33,86].
IL-38 is a cytokine of the IL-1 family that regulates the NOD-like receptor thermal protein domain associated protein 3 (NLRP3)/IL-1β signaling pathway, inhibits the activation of NLRP3 inflammasomes, enhances CD4 CD25 Treg cell immune function, reduces the expression of inflammatory mediators and pro-apoptotic proteins, attenuates early inflammatory response and apoptosis, improves the depletion of effector T cells during sepsis, and increases the survival rate of septic mice (>60%) [34,35]. FGF-15 is an analog of FGF-19, which significantly improves liver inflammation and apoptosis in septic mice by targeting fibroblast growth factor receptor 4, reducing the proportion of Tregs, and increasing the survival rate of septic mice by 40% [36]. Insulin-like growth factor 1, which is an anti-apoptosis factor, reduces hippocampal apoptosis and improves cognitive impairment by inhibiting the overexpression of cytochrome c and tumor necrosis factor receptors in a rat model of septic encephalopathy (SE) [38]. Maf1 is a transcriptional regulator of RNA polymerase III that reduces inflammation and apoptosis by competitively binding to the NLRP3 and inhibiting the expression and activity of NLRP3. Maf1 could also be used to prevent sepsis-associated encephalopathy (SAE) [39]. Ac2-26 is an N-terminal active peptide of Annexin A1 that targets formyl peptide receptor-2 receptors and inhibits PI3K and AKT phosphorylation. Ac2-26 downregulates the level of NF-κB, reduces the expression of inflammatory cytokines and pro-apoptotic proteins in septic mice, inhibits inflammation and apoptosis, improves renal injury, and increases the 7-day survival rate of CLP mice (>40%) [40].
Vaspin (Serpin A12) is an adipose factor of the Serpin family that inhibits the expression of kallikrein 7; downregulates the expression of pro-apoptotic protein Bcl2-associated X (bax), TNF-α, and other inflammatory factors; promotes the expression of anti-apoptotic protein bcl-2; reduces inflammatory response and apoptosis; and improves septic heart injury in mice. Vaspin treatment increased the survival rate of CLP mice by 50% [41].
Heat shock protein 22 (Hsp22/HspB8) functions as a molecular chaperone and an anti-inflammatory and anti-oxidation molecule. Hsp22 may inhibit the expression of apoptotic proteins and inflammatory factors by promoting AMPK activation and reducing mTOR phosphorylation levels, and has been shown to significantly improve myocardial injury and inflammation in an LPS-induced sepsis mouse model [37].
APN, secreted by the adipocytes, activates the PI3K/AKT pathway, reduces the proportion of the apoptotic proteins bax and cleaved-caspase-3 induced by LPS, and improves cardiomyocyte apoptosis induced by sepsis. In addition, APN inhibits mTOR activation by promoting AMPK phosphorylation, regulating the AMPK/mTOR pathway, and reducing inflammatory responses and liver injury in septic rats [42,43].
Irisin is a polypeptide hormone produced by the cleavage of membrane protein fibronectin type III domain-containing protein 5. Irisin can decrease the IL-1 β and Gasdermin D (GSDMD) levels by regulating the mitochondrial ubiquitin ligase (MITOL)/GSDMD pathway, improving the levels of LDH and creatine kinase-MB, and reducing inflammation and cardiomyocyte scorch caused by sepsis [44].
(2) The abbreviations have been reviewed again, and revised as necessary. Revisions in the text are highlighted by the utilization of the color red.
- The paper is much improved, it needs only minor english checks.
Response:
We would like to express our profound appreciation to the editors for their invaluable suggestions. The manuscript underwent editing by Editage, a professional language editing company.
